# Knowing When to Quit: A Principled Framework for Dynamic Abstention in LLM Reasoning

Hen Davidov [1]    Nachshon Cohen [2]    Oren Kalinsky [2]    Yaron Fairstein [2]    Guy Kushilevitz [2]    Ram Yazdi [2]
Patrick Rebeschini [1] [2]

## Abstract

LLMs utilizing chain-of-thought reasoning often waste substantial compute by producing long, incorrect responses. Abstention can mitigate this by withholding outputs unlikely to be correct. While most abstention methods decide to withhold outputs before or after generation, dynamic mid-generation abstention considers early termination of unpromising reasoning traces at each token position. Prior work has explored empirical variants of this idea, but principled guidance for the abstention rule remains lacking. We present a formal analysis of dynamic abstention for LLMs, modeling abstention as an explicit action within a regularized reinforcement learning framework. An abstention reward parameter controls the trade-off between compute and information. We show that abstaining when the value function falls below this reward strictly outperforms natural baselines under general conditions. We further derive a principled and efficient method to approximate the value function. Empirical results on mathematical reasoning and toxicity avoidance tasks support our theory and demonstrate improved selective accuracy over existing methods.

## 1. Introduction

LLMs have demonstrated remarkable proficiency across a broad spectrum of reasoning-intensive domains, including mathematical problem solving, scientific question answering, program synthesis, and symbolic logic (OpenAI et al., 2023; Yang et al., 2025). A primary catalyst for these advancements is their capacity to generate long-form natural language responses that explicitly articulate intermediate

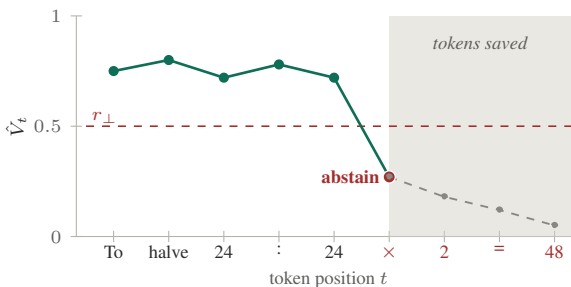

*Figure 1.* Dynamic value-thresholding. We estimate the value function of the current prefix with $\hat{V}_t$. When $\hat{V}_t$ falls below the abstention reward $r_\perp$, generation halts; the shaded region marks the tokens saved relative to the would-be (incorrect) continuation.

reasoning steps, a paradigm best exemplified by Chain-of-Thought (CoT) prompting (Kojima et al., 2022; Wei et al., 2022). However, while such verbose reasoning substantially enhances task performance, it increases the computational cost of inference.

Crucially, this computational expenditure does not guarantee accuracy. Models frequently generate incorrect responses while still incurring the full latency and cost of long-form generation. This inefficiency is compounded by the empirical observation that incorrect reasoning traces tend to be longer than correct ones (Janiak et al., 2025; Zhao et al., 2025). This creates two distinct problems: incorrect answers are delivered to users, and significant computational resources are wasted generating them.

A natural solution is to allow abstention: withholding low-quality outputs or deferring to a fallback mechanism such as a human expert or a more capable model (Wen et al., 2024). The landscape of abstention methods (surveyed more broadly in Appendix A) can be understood through the lens of *when* the abstention decision is made, and how well each approach addresses the two problems above.

At one extreme lies **output processing**: deciding whether to abstain after the complete response has been generated (Kadavath et al., 2022; Kuhn et al., 2023; Azaria & Mitchell, 2023). We focus on reasoning tasks with verifiable rewards: domains such as mathematical problem-solving, code generation, and symbolic reasoning where correctness can be

[1]University of Oxford. Work done partly during an internship at amazon. [2]Amazon. Correspondence to: Hen Davidov <hen.davidov@stats.ox.ac.uk>.

*Proceedings of the 43rd International Conference on Machine Learning*, Seoul, South Korea. PMLR 306, 2026. Copyright 2026 by the author(s).

checked deterministically. In these settings, the challenge is not whether we can identify errors, but whether we can do so *before* expending resources on full CoT generation.

At the opposite extreme, **input processing** attempts to predict failure based solely on the prompt (Kadavath et al., 2022; Zhang et al., 2024a; Cheng et al., 2024). This approach addresses the efficiency problem effectively - if abstention occurs, no computation is wasted. However, input processing may have limited accuracy in selecting which prompts to abstain from answering (Kadavath et al., 2022). Even under deterministic decoding, the prompt alone oftentimes provides little information about which specific reasoning path the model will take or where it might fail. This limitation is compounded when LLMs are deployed with stochastic sampling, a common practice that improves performance on reasoning tasks (Hochlehnert et al., 2025).

Recently, Afzal et al. (2025) and Zhang et al. (2025) proposed observing partial generations and making abstention decisions at **fixed token positions**. While observing part of the output improves upon input-only methods, the point at which a trace's fate becomes clear varies across problems. Zhang et al. (2025) address this with an empirical dynamic rule that can terminate generation at varying token positions. While their technique demonstrated empirical promise, it requires segmenting traces into reasoning chunks and using an external model to label intermediate answer correctness. However, a correct intermediate answer may later be revised incorrectly, and vice versa. What matters for abstention is whether the final answer will be correct.

To directly target final-answer correctness, we work within the KL-regularized reinforcement learning framework (Jaques et al., 2017; Ziegler et al., 2019). Since correctness is verified only for the final answer, we model rewards as sparse and binary. The reward for a finalized answer is $1$ if it is correct, and $0$ otherwise. To accommodate abstention, we augment the action space with an abstention token $\perp$ and associate it with an abstention reward $r_\perp$. This $r_\perp$ quantifies the utility of abstention, whether that means deferral to a human expert or routing to a more capable model.

Within this framework, we analyze a natural decision rule, **dynamic value-thresholding**, illustrated in Figure 1. We abstain when the value function falls below $r_\perp$. In our sparse reward setting, the value function reduces to the expected terminal reward from the current state, giving it a natural interpretation as the model's confidence in eventual success. If this falls below the fallback reward, continuing is wasteful. This threshold $r_\perp$ gives practitioners direct control over the abstention rate: lower $r_\perp$ yields higher selective accuracy, while higher $r_\perp$ yields greater compute savings.

We formalize when and why this rule outperforms alternatives. Throughout, we say a policy dominates another

policy if it achieves at least the same expected reward, and it *strictly dominates* a policy if it achieves strictly higher reward. We obtain the following results, assuming perfect knowledge of the value function:

- **Dominance over no abstention** (Proposition 4.2): Given any base LLM, dynamic value-thresholding dominates non-abstaining. it strictly dominates if abstention occurs with positive probability.

- **Dominance over fixed-position abstention** (Corollaries 4.5 and 4.6): Given any base LLM, dynamic value-thresholding dominates input processing based on value-thresholding. Further, it dominates fixed-position abstention at position $k > 1$ when low-value states cannot recover before $k$. Finally, dominance is strict if the dynamic rule has a positive probability of abstaining after the $k$-th token position.

- **Optimality** (Proposition 4.7): Given a base LLM that dominates all non-abstaining LLMs, and assuming zero KL regularization, dynamic value-thresholding dominates all abstention-enabled policies.

- **Improvement bound** (Proposition 4.9): Given any base LLM, and assuming zero KL regularization and Lipschitz continuity, dynamic abstention improves over no abstention at a rate that is upper-bounded linearly in the abstention rate.

We address the estimation of the value function in Section 5. We show that for binary rewards and zero KL regularization, the value function equals the probability of trajectory success conditioned on the current state. As a result, parallel binary cross-entropy training at all non-terminal token positions recovers this quantity under realizability, as proven by Proposition B.2. In Appendix B.8 we extend this result to general bounded rewards using an MSE loss.

We utilize the binary cross entropy loss to train a two-layer MLP probe on the LLM's hidden states. Compared to the 7B and 4B-parameter models tested in Section 6, the two-layer MLP adds roughly $10^4$ parameters, making probe inference and training overhead negligible.

In Section 6, we measure the performance of dynamic value-thresholding compared to natural baselines, examine robustness to imperfect value estimation, show cross-dataset generalization of our value estimators, and validate our theoretical findings. We perform our experiments on two LLMs: Qwen2.5 (Qwen et al., 2025) and Phi-3 (Abdin et al., 2024), and two chain-of-thought math reasoning tasks: GSM8K (Cobbe et al., 2021) and OlympiadBench (Sun et al., 2025). Dynamic value-thresholding outperforms all tested baselines in both expected reward and selective accuracy across the full range of abstention rates. The advantage is most

pronounced on harder problems: on OlympiadBench, where the base Phi-3 achieves only 16% baseline accuracy, dynamic abstention reaches 64% selective accuracy at 90% abstention, nearly double the 34% achieved by baseline methods. For Qwen on OlympiadBench (43% baseline), dynamic abstention achieves 91% selective accuracy at 90% abstention compared to 75% for the best baseline. These gains come with modest efficiency trade-offs: dynamic abstention retains 63–86% of the token savings of input-only methods at 10% abstention rates, rising to 92–95% at 90% abstention rates. Code is available at `our repo`.

## 2. MDP Formulation

We formulate text generation as a Markov Decision Process (MDP) with sparse, binary rewards. Given a prompt $x$, we define a response $y$ as a sequence of tokens $[y_1, y_2, \ldots, y_k]$ drawn from a vocabulary $\mathcal{V}$, where the sequence length $k$ is bounded by a maximum $T$. Let $y_{1:t} := [y_1, \ldots, y_t]$ denote a prefix of length $t$. The empty sequence is denoted as $y_{1:0}$. At any step $t$, the language model acts as a policy $\pi$, inducing a probability distribution over the next token, $\pi(y_t \mid x, y_{1:t-1})$. For simplicity, we also denote the probability distribution over full responses induced by $\pi$ as $\pi(y|x)$.

To align the model using a sequence-level reward signal, we formulate the MDP with a sparse, token-wise reward. This reward is zero for all intermediate steps and non-zero only upon the generation of the end-of-sequence token (`eos`):

$$R_\beta(x, y_{1:t}; \pi) \tag{1}$$
$$= \begin{cases} r(x, y_{1:t}) - \beta \log \frac{\pi(y_{1:t}|x)}{\pi_{\mathrm{ref}}(y_{1:t}|x)}, & \text{if } y_t = \texttt{eos}, \\ 0, & \text{otherwise.} \end{cases}$$

where $r(x, y)$ is the terminal reward function (e.g., a verifier score), $\pi_{\mathrm{ref}}$ is the reference policy (typically the supervised fine-tuned model), and $\beta > 0$ is the KL penalty coefficient.

For a complete response $y$ terminating in `eos`, we denote the total reward by

$$R_\beta(x, y; \pi) = \sum_{t=1}^{T} R_\beta(x, y_{1:t}; \pi). \tag{2}$$

This is the sum of token-level rewards along the trajectory, which collapses to a single term at the terminal token due to the sparse reward structure.

Let $\rho$ be the prompt distribution. The optimization objective $J_\beta(\pi)$ is the expected total reward:

$$J_\beta(\pi) = \mathbb{E}_{x \sim \rho, y \sim \pi(\cdot|x)} \left[ R_\beta(x, y; \pi) \right]. \tag{3}$$

An optimal policy $\pi_\beta^*$ is then a policy such that $\pi_\beta^* \in \mathrm{argmax}_{\pi \in \Pi_\mathcal{V}} J_\beta(\pi)$, where $\Pi_\mathcal{V}$ is the set of all policies mapping states to the probability simplex over $\mathcal{V}$.

Analogously, we define the state value function $V_\beta(x, y_{1:t}; \pi)$ as the expected future reward from state $(x, y_{1:t})$:

$$V_\beta(x, y_{1:t}; \pi) = \mathbb{E}_{y \sim \pi(\cdot|x, y_{1:t})} \left[ \sum_{k=0}^{T-t} R_\beta(x, y_{1:t+k}; \pi) \right].$$

## 3. Formulating Abstention

We augment the MDP action space to include a special abstention action, denoted by $\perp$. An augmented policy $\pi^\dagger$ selects tokens from $\mathcal{V}^\dagger = \mathcal{V} \cup \{\perp\}$. This token serves as a terminal action: if the policy selects $y_t = \perp$, generation halts immediately, and a fallback mechanism, i.e. deferral or refusal, is triggered.

To accommodate abstention, we extend the reward formulation in two ways. First, we assign a fixed scalar reward $r_\perp \in (0, 1]$ to any trajectory terminating in $\perp$. This reward quantifies the utility of the fallback and serves as a tunable hyperparameter controlling the abstention rate.

Second, we modify the KL term. Since the reference policy $\pi_{\mathrm{ref}}$ assigns zero probability to $\perp$, penalizing deviation for this action is undefined. We therefore compute KL divergence only over non-abstention tokens, using the *abstention-stripped policy* $s(\pi^\dagger)$, the distribution of $\pi^\dagger$ conditioned on not abstaining in the next token:

$$s(\pi^\dagger)(y_t \mid x, y_{1:t-1}) = \frac{\pi^\dagger(y_t \mid x, y_{1:t-1})}{1 - \pi^\dagger(\perp \mid x, y_{1:t-1})}, \qquad y_t \in \mathcal{V}.$$

We utilize these changes to extend the token-level reward of Equation (1) to abstention actions:

$$R_\beta(x, y_{1:t}; \pi^\dagger) \tag{4}$$
$$= \begin{cases} r_\perp, & \text{if } y_t = \perp, \\ r(x, y_{1:t}) - \beta \log \frac{s(\pi^\dagger)(y_{1:t} \mid x)}{\pi_{\mathrm{ref}}(y_{1:t} \mid x)}, & \text{if } y_t = \texttt{eos}, \\ 0, & \text{otherwise.} \end{cases}$$

$s(\pi^\dagger)$ is well-defined only when $\pi^\dagger(\perp \mid x, y_{1:t-1}) < 1$. However, since $s(\pi^\dagger)$ appears only in the `eos` case, which requires reaching termination without abstaining, the undefined case never arises in computing rewards. For any policy $\pi$ that never abstains, this reduces to the standard reward in Equation (1), so the formulation is a strict generalization. We extend $V_\beta$ and $J_\beta$ accordingly.

### 3.1. Understanding $J_\beta$

The objective $J_\beta$ admits a natural decomposition. Let

$$\alpha = \mathbb{P}_{x \sim \rho, y \sim \pi^\dagger(\cdot|x)}(\exists t : y_t = \perp) \tag{5}$$

denote the abstention rate, and let $S = \mathbb{E}_{x \sim \rho, y \sim \pi^\dagger(\cdot|x)}[R_\beta(x, y; \pi^\dagger) \mid \texttt{eos} \in y]$ denote the

expected reward among non-abstained responses. By the law of total expectation:

$$J_\beta(\pi^\dagger) = \alpha \cdot r_\perp + (1 - \alpha) \cdot S. \tag{6}$$

Rewriting as $J_\beta = S - \alpha(S - r_\perp)$ shows that maximizing $J_\beta$ is equivalent to maximizing $S$ with an abstention penalty of $(S - r_\perp)$; varying $r_\perp$ traces out different points on the accuracy-abstention frontier. When $\beta = 0$ and $r(x, y) \in \{0, 1\}$ indicates correctness, $S$ reduces to expected selective accuracy: $\mathbb{P}(\text{correct} \mid \text{not abstain})$.

This connects $J_\beta$ to selective classification, formalized by El-Yaniv & Wiener (2010), where the goal is to maximize expected selective accuracy subject to an abstention budget $\alpha_0$. In the non-sequential classification setting, Geifman & El-Yaniv (2017) proposed a solution for black-box classifiers. Sequential generation introduces a fundamental difficulty: the constraint aggregates across all token positions, yet abstention decisions must be made locally at each position.

Maximizing $J_\beta$ offers an alternative that circumvents this difficulty. By Bellman's principle of optimality (Bellman, 1952), a policy is globally optimal if and only if its value function is optimal at every reachable state. We now use this to derive a necessary condition that any optimal abstention policy must satisfy.

Consider the value function at state $(x, y_{1:t-1})$, and let $p = \pi^\dagger(\perp \mid x, y_{1:t-1})$ denote the abstention probability. Since the value is linear in $p$:

$$V_\beta(x, y_{1:t-1}; \pi^\dagger)$$
$$= p \cdot r_\perp + (1 - p) \cdot \mathbb{E}\left[R_\beta(x, y_{1:t}; \pi^\dagger) + V_\beta(x, y_{1:t}; \pi^\dagger)\right],$$

where the expectation is taken over $y_t \sim s(\pi^\dagger)(\cdot | x, y_{1:t-1})$, which, when $p < 1$, is the distribution conditioned on the next action not being abstention. The value function in the expectation is defined using the full policy $\pi^\dagger$, as the policy might abstain at a later token position. Maximizing over $p$ yields a deterministic rule: the optimal policy abstains if and only if

$$\mathbb{E}\left[R_\beta(x, y_{1:t}; \pi^\dagger) + V_\beta(x, y_{1:t}; \pi^\dagger)\right] < r_\perp. \tag{7}$$

The global abstention rate–accuracy tradeoff thus reduces to a local comparison at each state: abstain whenever the expected value of continuing falls below the fallback reward. However, Equation (7) is a necessary condition for optimality rather than a directly implementable rule. The condition depends on the value function of $\pi^\dagger$, but modifying the abstention behavior to satisfy the condition changes $\pi^\dagger$, which in turn changes the value function.

## 3.2. A Natural Dynamic Abstention Rule

We resolve this circularity by constructing an augmented policy $a(\pi)$ that satisfies Equation (7). Define $a(\pi)$ to abstain at state $(x, y_{1:t-1})$ if and only if $V_\beta(x, y_{1:t-1}; \pi) < r_\perp$, and otherwise follow the distribution of $\pi$. Formally, letting $b = \mathbb{I}\{V_\beta(x, y_{1:t-1}; \pi) < r_\perp\}$, the **dynamic value-thresholding** policy is

$$a(\pi)(y_t \mid x, y_{1:t-1}) \tag{8}$$
$$= \begin{cases} b, & \text{if } y_t = \perp, \\ (1 - b)\pi(y_t \mid x, y_{1:t-1}), & \text{otherwise.} \end{cases}$$

The intuition is straightforward: if the expected future reward falls below the fallback utility, continuing generation is wasteful. The threshold $r_\perp$ provides direct control: lower $r_\perp$ yields higher selective accuracy, while higher $r_\perp$ yields greater compute savings. We verify in Appendix B.1 that $a(\pi)$ satisfies the necessary condition in Equation (7). The following section establishes that this rule is optimal under idealized conditions and beneficial more generally.

# 4. Theoretical Guarantees

We establish that dynamic abstention improves over natural baselines, characterize a setting in which it is optimal, and bound the magnitude of improvement. All results assume access to an oracle value function; Section 5 addresses estimation.

The foundation for our analysis is that $a(\pi)$ achieves *value dominance*: its value function everywhere dominates both the fallback reward and the base policy's value.

**Lemma 4.1** (Value Dominance). *For all reachable, non-terminal states $(x, y_{1:t})$:*

$$V_\beta(x, y_{1:t}; a(\pi)) \geq \max\left(r_\perp, V_\beta(x, y_{1:t}; \pi)\right). \tag{9}$$

*Moreover, the inequality $V_\beta(x, y_{1:t}; a(\pi)) > V_\beta(x, y_{1:t}; \pi)$ is strict if and only if some state reachable from $(x, y_{1:t})$ under $\pi$ satisfies $V_\beta(x, y_{1:t}; \pi) < r_\perp$.*

Intuitively, $a(\pi)$ intervenes precisely when intervention helps, replacing low-value continuations with $r_\perp$ while leaving high-value trajectories untouched. The formal proof (Appendix B.2) proceeds by backward induction.

## 4.1. Improvement Over the Base Policy

We first show that $a(\pi)$ does not deteriorate the performance of *any* base policy $\pi$, with strict improvement whenever abstention occurs with positive probability.

**Proposition 4.2** (Dominance Over No Abstention). *Let $\pi$ be any policy over $\mathcal{V}$, and let $\beta \geq 0$. Then:*

$$J_\beta(a(\pi)) \geq J_\beta(\pi) \quad \text{and} \quad J_\beta(a(\pi)) \geq r_\perp. \tag{10}$$

*Moreover, $J_\beta(a(\pi)) > J_\beta(\pi)$ if and only if there exists a reachable, non-terminal state $(x, y_{1:t})$ with $V_\beta(x, y_{1:t}; \pi) < r_\perp$.*

*Proof.* Since $J_\beta(\pi) = \mathbb{E}_{x\sim\rho}[V_\beta(x, y_{1:0}; \pi)]$ and $J_\beta(a(\pi)) = \mathbb{E}_{x\sim\rho}[V_\beta(x, y_{1:0}; a(\pi))]$, both inequalities follow directly from Lemma 4.1. The strict inequality condition follows from the second part of the lemma, also applied at $t = 0$. □

In practice, reasoning models often embark on solution paths that become unrecoverable partway through generation. In these states $V_\beta < r_\perp$, triggering abstention and guaranteeing strict improvement.

### 4.2. Comparison with Fixed-Position Abstention

We now compare dynamic value-thresholding to approaches that make the abstention decision at a single, predetermined token position.

**Definition 4.3** (Fixed-Position Abstention). *For $k \in \{1, \dots, T\}$, let $f(\pi; k)$ denote the policy that applies value-based abstention only at timestep $k$ and follows $\pi$ otherwise:*

$$f(\pi; k)(y_t | x, y_{1:t-1}) = \begin{cases} a(\pi)(y_t | x, y_{1:t-1}), & \text{if } t = k, \\ \pi(y_t | x, y_{1:t-1}), & \text{otherwise.} \end{cases}$$

This framework captures several existing approaches. Input processing corresponds to $f(\pi; 1)$, while mid-generation fixed-position approaches correspond to $f(\pi; k)$ for some $1 < k < T$. In Appendix B.3 we prove the following results.

**Proposition 4.4** (Dominance over Fixed-Position Abstention). *Let $k \in \{1, \dots, T\}$ and $\beta \geq 0$. Define the first abstention time under $a(\pi)$:*

$$\tau = \min\{t \geq 1 : V_\beta(x, y_{1:t-1}; \pi) < r_\perp\},$$

*with $\tau = \infty$ if no such $t$ exists. Then:*

$$\begin{aligned} J_\beta(a(\pi)) \geq & J_\beta(f(\pi; k)) \quad\quad\quad\quad (11) \\ & - \mathbb{E}\left[\mathbb{I}\{\tau < k\} \cdot (V_\beta(x, y_{1:k-1}; \pi) - r_\perp)^+\right], \end{aligned}$$

*where the expectation is over $x \sim \rho$ and $y_{1:k-1} \sim \pi(\cdot \mid x)$.*

*Moreover, strict inequality holds in* (11) *if and only if there exists a reachable, non-terminal state $(x, y_{1:t})$ such that:*

1. *$t \geq k$,*

2. *$V_\beta(x, y_{1:t}; \pi) < r_\perp$, and*

3. *$V_\beta(x, y_{1:s}; \pi) \geq r_\perp$ for all $s < k$.*

The term $\mathbb{E}[\mathbb{I}\{\tau < k\} \cdot (V_\beta(x, y_{1:k-1}; \pi) - r_\perp)^+]$ is the *expected recovery surplus*: it is nonzero only when $a(\pi)$ abstains before position $k$ (i.e., $\tau < k$) and the trajectory subsequently recovers to have value above $r_\perp$ at position $k - 1$. This captures the "option value of waiting" that $f(\pi; k)$ exploits by delaying its abstention decision. The following corollaries identify conditions under which this term vanishes.

**Corollary 4.5.** *If $\mathbb{P}(\tau < k) = 0$, then $J_\beta(a(\pi)) \geq J_\beta(f(\pi; k))$, with strict inequality as characterized in Proposition 4.4.*

**Corollary 4.6.** *Suppose low-value states cannot recover at position $k$: for all $t < k - 1$, if $V_\beta(x, y_{1:t}; \pi) < r_\perp$, then $V_\beta(x, y_{1:k-1}; \pi) < r_\perp$ for every $y_{t+1:k-1}$ in the support of $\pi(\cdot \mid x, y_{1:t})$. Then $J_\beta(a(\pi)) \geq J_\beta(f(\pi; k))$, with strict inequality as characterized in Proposition 4.4.*

Corollary 4.5 applies directly to input processing baselines ($k = 1$), where $\tau < 1$ is impossible by definition. Corollary 4.6 applies when early errors constrain future paths, and no continuation can restore the value at position $k - 1$ above $r_\perp$.

### 4.3. Optimality Under An Optimal Base Policy

The preceding results hold for any $\beta \geq 0$. For the remainder of this section, we focus on $\beta = 0$. While LLMs are trained with $\beta > 0$ to prevent mode collapse, at inference time practitioners care about correctness, not proximity to a reference distribution. This setting also admits cleaner characterizations.

We now ask: when is dynamic value-thresholding globally optimal? Beyond $\beta = 0$, an optimal base policy $\pi_0^* \in \operatorname{argmax}_\pi J_0(\pi)$ is required. This reflects a fundamental limitation. Abstention can avoid bad outcomes but it cannot improve the quality of completed generations. Under these conditions, dynamic value-thresholding achieves the optimal objective over all abstention-augmented policies.

**Proposition 4.7** (Optimality of Dynamic Abstention). *Let $\beta = 0$, and let $\pi_0^* \in \operatorname{argmax}_{\pi \in \Pi_V} J_0(\pi)$ be an optimal policy over the original vocabulary $\mathcal{V}$. Then $a(\pi_0^*)$ is optimal among all policies over the augmented vocabulary $\mathcal{V}^\dagger$:*

$$a(\pi_0^*) \in \operatorname*{argmax}_{\pi^\dagger \in \Pi_{\mathcal{V}^\dagger}} J_0(\pi^\dagger).$$

The proof is detailed in Appendix B.4.

### 4.4. Characterizing the Magnitude of Improvement

The magnitude of improvement depends on how gradually the value function changes during generation. Improvement accrues at the moment $V_0$ first crosses below $r_\perp$: the gain at that step is $r_\perp - V_0$, which cannot exceed the single-step

change in $V_0$, since the value one step earlier was still at least $r_\perp$. We formalize this via a Lipschitz condition.

**Definition 4.8** (Value Function Lipschitzness). *A policy $\pi$ has an $L$-Lipschitz value function if for all reachable states $(x, y_{1:t})$ with $1 \le t \le T$:*

$$|V_0(x, y_{1:t-1}; \pi) - V_0(x, y_{1:t}; \pi)| \le L.$$

For binary rewards $r(x, y) \in \{0, 1\}$ the condition holds trivially with $L = 1$; tighter constants arise when a single token does not determine the final outcome. This is plausible for chain-of-thought reasoning where models self-correct via phrases like "wait" or "let me reconsider" (Wei et al., 2022).

The Lipschitz constant bounds improvement after generation has started. When abstention occurs immediately at $t = 1$, there is no preceding state and the gain is instead bounded by $r_\perp$. The following result formalizes this insight.

**Proposition 4.9** (Linear Improvement Bound). *Let $\beta = 0$ and $r(x, y) \in \{0, 1\}$. Suppose $\pi$ has an $L$-Lipschitz value function. Let $\alpha$ be the abstention rate (Equation (5)), and let $\alpha_1 = \mathbb{P}_{x \sim \rho}(V_0(x, y_{1:0}; \pi) < r_\perp)$ be the probability of immediate abstention. Then:*

$$J_0(a(\pi)) - J_0(\pi) \le \alpha_1 \cdot r_\perp + (\alpha - \alpha_1) \cdot L.$$

See Appendix B.5 for the proof.

## 5. Value Approximation

We now address the estimation of $V_\beta(x, y_{1:t}; \pi)$ in practice. Specifically, we parameterize the value estimator as an MLP probe on hidden states, and prove that binary cross-entropy recovers the value function under a realizability assumption.

Estimating the value function generally requires predicting expected cumulative future rewards, which vary by position. The sparse reward structure simplifies this considerably: since all reward is concentrated at the terminal token, the value function at any non-terminal state reduces to the expected terminal reward. This is the same target for every prefix of a trajectory, enabling value estimation through parallel supervised learning across prefixes of completed outputs. This has been previously utilized for related methods, including Han et al. (2024); Mudgal et al. (2024); Snell et al. (2023); Tang et al. (2023); Wang et al. (2016).

For the reasoning tasks we consider, rewards are binary correctness indicators $r(x, y) \in \{0, 1\}$. Under this condition and $\beta = 0$, the value function at any non-terminal state $(x, y_{1:t})$ reduces to the conditional probability of correctness:

$$V_0(x, y_{1:t}; \pi) = \mathbb{P}_\pi(r(x, y) = 1 \mid x, y_{1:t}). \quad (12)$$

For terminal states ($\text{eos} \in y_{1:t}$), $V_0(x, y_{1:t}; \pi) = 0$ since no future reward can be collected. A formal derivation from the general $\beta \ge 0$ case appears in Appendix B.6. Equation (12) is significant because it reduces value estimation to a binary classification problem. This is precisely the quantity that binary cross-entropy is designed to recover.

We parameterize the estimator as a family of predictors $\{\hat{V}_t\}_\theta$ sharing parameters $\theta$. For non-terminal states ($\text{eos} \notin y_{1:t}$), we define $\hat{V}_t(x, y_{1:t}; \theta) = \text{MLP}_\theta(h_t)$, where $h_t$ is the hidden state at position $t$. The causal structure of the transformer ensures $h_t$ depends only on $(x, y_{1:t})$, so the estimator has access to exactly the conditioning information in Equation (12). For terminal states ($\text{eos} \in y_{1:t}$), we set $\hat{V}_t(x, y_{1:t}; \theta) = 0$ by definition, matching the structure of the true value function.

Let $\mathcal{D} = \{(x_i, y_i)\}_{i=1}^N$ be a dataset of completed trajectories where $x_i \sim \rho$ and $y_i \sim \pi(\cdot | x_i)$. Since the reward is the same binary label $r(x, y)$ for every prefix of a trajectory, we train all positions in parallel using binary cross-entropy:

$$\mathcal{L}(\theta) = \mathbb{E}_{(x,y) \sim \mathcal{D}} \left[ \sum_{t=0}^{c-1} \ell \left( \hat{V}_t(x, y_{1:t}; \theta), r(x, y) \right) \right] \quad (13)$$

where $\ell(\hat{p}, r) = -r \log \hat{p} - (1 - r) \log(1 - \hat{p})$ and the sum runs over non-terminal positions only.

Under a standard realizability assumption, $V_t \in \{\hat{V}_t\}_\theta$, minimizing this objective recovers the true value function. A formal statement and proof appear in Proposition B.2 (Appendix B.7). For continuous rewards or $\beta > 0$, mean squared error provides an analogous guarantee (Proposition B.3, Appendix B.8).

This approach requires only hidden state extraction and probe training, and adds negligible inference overhead.

Pseudocode for training the value estimator and for inference is given in Algorithms 1 and 2, respectively.

## 6. Experiments

We evaluate dynamic value-thresholding on two tasks: chain-of-thought mathematical reasoning (Sections 6.1–6.4) and toxicity avoidance (Section 6.5). The mathematical reasoning experiments form the bulk of the evaluation, and the remainder of this overview describes their setup; details for the toxicity experiment are deferred to Section 6.5.

Within mathematical reasoning, Section 6.1 compares methods on selective accuracy. Section 6.2 validates the dominance and improvement guarantees from Section 4. Section 6.3 tests whether the value estimator generalizes across datasets without retraining. Section 6.4 examines the sensitivity of our method to imperfect value estimation.

**Models and Datasets.** We evaluate on two LLMs: Qwen2.5-7B-Instruct (Qwen et al., 2025) and Phi-3-mini-4k-Instruct (Abdin et al., 2024), across two chain-of-thought mathematical reasoning benchmarks: GSM8K (Cobbe et al., 2021) and OlympiadBench (Sun et al., 2025). GSM8K contains grade-school math problems where both models achieve high baseline accuracy ($87\%$ for Phi-3 and $88\%$ for Qwen), while OlympiadBench contains competition-level problems where baseline accuracy is substantially lower ($16\%$ for Phi-3 and $43\%$ for Qwen).

**Value Function Estimation.** Following Section 5, we train an MLP probe on hidden states to estimate $V_0(x, y_{1:t}; \pi)$. The probe is a two-layer MLP trained on the final layer hidden states, using binary cross-entropy loss as described in Equation (13). To train the dynamic abstention method, for each model-dataset pair, we generate full trajectories with the base model and extract hidden states from the final transformer layer.

**Baselines.** We consider three baseline estimation methods for the abstention decision, each evaluated at two token positions: $t = 0$ (input-processing) and $t = k$ (fixed-position mid-generation), corresponding to $f(\pi; 1)$ and $f(\pi; k)$ in Definition 4.3.

- **Constant Step Probe**: A two-layer MLP trained on hidden states to predict answer correctness via binary cross-entropy, evaluated at a fixed step $t$ (Kadavath et al., 2022; Afzal et al., 2025).

- **Self-Assessment**: The model is prompted with "Can you correctly answer this question?" and the logits for the `yes`/`no` tokens are used for classification (Zhang et al., 2024b).

- **LoRA Abstention**: The base LLM is finetuned via LoRA with explicit `abstain`/`do not abstain` tokens whose logits are used for classification (Zhang et al., 2024a).

We also report the base policy $\pi$ without abstention, denoted **no abstention**. Our dynamic method uses the same Constant Step Probe architecture but evaluates it at every token position, abstaining when the estimate falls below $r_\perp$.

We set $k = 20$ for GSM8K and $k = 100$ for Olympiad-Bench, reflecting the longer reasoning traces on harder problems. Both choices are arbitrary: the optimal position varies across datasets, models, and desired abstention rates, and there is no principled data-independent criterion for selecting $k$ (Appendix K). This arbitrariness is itself an argument for the dynamic approach, which adapts its stopping point to each trace.

Further implementation and experimental setup details are found in Appendix H. All reported results are averaged over 5 random seeds (seeds 42–46), where each seed corresponds to an independent retraining of the Constant Step Probe; shaded regions in figures show $\pm 1$ standard deviation across seeds.

## 6.1. Selective Accuracy

We first evaluate methods on selective accuracy: the mean answer correctness among non-abstained samples. This metric isolates each method's ability to *rank* samples by correctness probability.

**Results.** Figure 2 shows selective accuracy versus abstention rate across all model–dataset combinations, comparing dynamic abstention against both input-processing baselines ($t = 0$) and fixed-position mid-generation baselines ($t = k$). Dynamic value-thresholding achieves the highest selective accuracy across all settings, abstention rates, and baseline categories.

Among input-processing baselines, the Constant Step Probe is the strongest competitor, while self-assessment and LoRA abstention show limited discrimination ability, often performing near or below the no-abstention baseline. Observing $k$ tokens of partial generation does not substantially help the self-assessment and LoRA methods: the fixed-position variants plateau near the no-abstention accuracy regardless of $\alpha$.

The advantage of dynamic value-thresholding over all baselines is most pronounced on harder problems and at higher abstention rates. On GSM8K, where baseline accuracy is already high, dynamic value-thresholding achieves selective accuracy of $0.92$ at $\alpha = 0.1$, rising to $0.99$ at $\alpha = 0.9$ for both models. On OlympiadBench, where the task is substantially harder, the gains are larger: on Phi-3 at $\alpha = 0.9$, dynamic abstention achieves $0.64$ selective accuracy, compared to $0.34$ for the best input-processing baseline and $0.33$ for the best fixed-position baseline — roughly double the performance of either family. On Qwen at $\alpha = 0.9$, dynamic abstention reaches $0.91$, compared to $0.69$ for the best input-processing baseline and $0.75$ for the best fixed-position baseline. The gap persists across all settings: at every abstention rate on every model–dataset combination, dynamic abstention strictly dominates the best baseline pointwise (Figure 2).

Appendix J reports the complementary precision metric $\mathbb{P}(\text{incorrect} \mid \text{abstained})$, confirming that abstentions are selectively targeted.

Figure 2 also plots the relative token savings of the dynamic method compared to the Constant Step Probe at $t = 0$. Dynamic mid-generation abstention achieves token savings

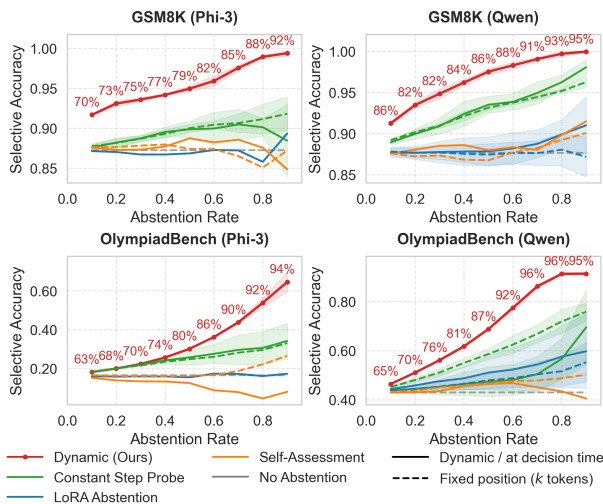

*Figure 2.* Selective accuracy versus abstention rate across all baselines: input-processing ($t = 0$), fixed-position mid-generation ($k = 20$ for GSM8K and $k = 100$ for Olympiad), and dynamic (ours). Token savings for dynamic abstention as a percent of constant step probing savings at $t = 0$ are labeled for each abstention rate.

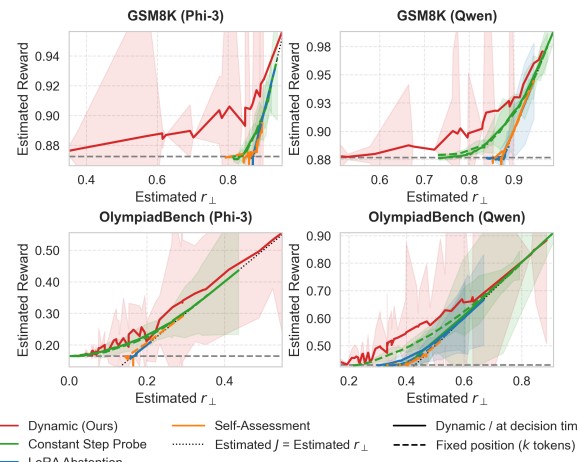

*Figure 3.* Estimated reward $\hat{J}$ versus calibrated $\hat{r}_\perp$ across all baselines. Proposition 4.2 predicts the curve lies above the diagonal (black dotted line) and no abstention (gray dashed line); Corollary 4.5 predicts dynamic (red) dominates all baselines at matched $\hat{r}_\perp$. The x-axis does not span $[0, 1]$ because $\hat{r}_\perp$ is determined by empirical accuracies at abstention boundaries; see Appendix D.3 for details.

close to the input-processing baselines but generally saves fewer tokens at low-to-mid abstention rates, with the gap narrowing at high abstention rates where dynamic abstention approaches parity with input-processing methods. Appendix F confirms that abstention occurs in the first half of generation on average, even at low abstention rates, with progressively earlier termination as $\alpha$ increases.

### 6.2. Reward Objective

Next, we evaluate methods on the reward objective under no KL regularization, $J_0$. Our theoretical results predict that, given an oracle value function, dynamic value-thresholding dominates both the base policy without abstention (Proposition 4.2) and input-only methods (Proposition 4.4). A stronger result, that dynamic value-thresholding is optimal among all abstention-enabled policies (Proposition 4.7), additionally requires that the base policy is itself optimal under $\beta = 0$. In practice, neither condition holds: we estimate the value function from data, and the base LLMs were trained with KL regularization and approximate optimization. Still, the results below provide empirical support for these predictions.

A subtlety arises in this evaluation. Our framework specifies $r_\perp$ as a threshold on the true value function, $V_0(x, y_{1:t}, \pi)$, but in practice we threshold our estimate $\hat{V}_t$ at some threshold $T_\alpha$. Because the estimate may be miscalibrated, $T_\alpha$ does not directly equal the effective $r_\perp$. The effective $r_\perp$ at threshold $T_\alpha$ is the true probability of correctness at the point where abstention is triggered, which may differ from $T_\alpha$. To address this, we use isotonic regression to estimate the transformation from estimated values to true correctness

probabilities. We apply this transformation on $T_\alpha$ to get a calibrated estimate of the effective $r_\perp$. We detail this methodology and the calibration analysis motivating it in Appendix D.

**Results.** Figure 3 plots estimated reward $\hat{J}$ against estimated $\hat{r}_\perp$ across abstention rates $\alpha \in \{0.02, 0.04, ..., 0.98\}$, comparing dynamic value-thresholding against all input-processing and fixed-position baselines. Consistent with Propositions 4.2, 4.4, and 4.7 the dynamic value-thresholding curve lies above the diagonal ($\hat{J} = \hat{r}_\perp$), and all baselines across all settings. In Appendix I, we show that improvement over no-abstention grows with the abstention rate, supporting the intuition behind Proposition 4.9.

### 6.3. Cross-Dataset Transfer

We evaluate whether the MLP probe generalizes across datasets without retraining. For each model, we train the probe on one dataset and evaluate it zero-shot on the other, using the in-domain threshold calibrated on a held-out split of the training dataset.

**Results.** Figure 4 shows selective accuracy for the transferred probe alongside the in-domain methods. Transfer generalizes well in both directions: the probe trained on GSM8K and evaluated on OlympiadBench closely matches in-domain dynamic performance across all abstention rates. The reverse direction — training on OlympiadBench and evaluating on GSM8K — shows a slightly larger gap. In both cases, the transferred probe consistently outperforms all baselines.

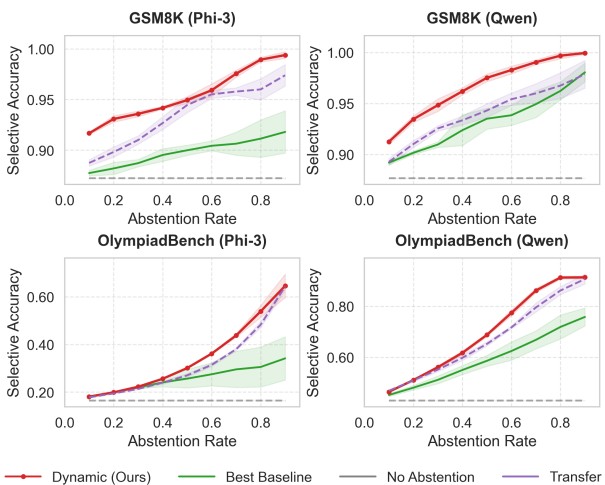

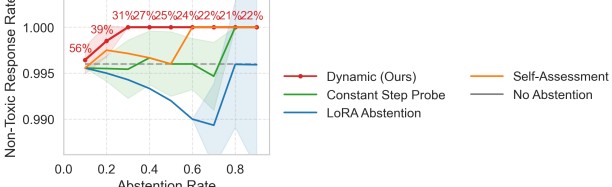

*Figure 5.* Non-toxic response rate among non-abstained samples versus abstention rate on RealToxicityPrompts (Qwen2.5-7B-Instruct). Red labels indicate token savings of the dynamic method relative to input-processing baselines.

*Figure 4.* Cross-dataset transfer: selective accuracy when the MLP probe is trained on one dataset and evaluated zero-shot on the other (purple dashed). The probe generalizes well, consistently outperforming all baselines in all settings. The best baseline (green) is chosen pointwise for each abstention rate, seed, and setting.

This suggests that rather than learning dataset-specific surface cues, it appears to capture general properties of the hidden state that are predictive of eventual correctness across problem types.

### 6.4. Robustness of the Value Estimator

The guarantees in Section 4 assume an oracle value function; in practice $\hat{V}$ differs from $V_0$. Three empirical checks show the method tolerates this.

**Threshold stability.** The abstention threshold $T_\alpha$ is calibrated on a held-out set. The achieved abstention rate tracks the target with mean absolute error below 1.2 percentage points across all settings (Appendix E).

**Ranking suffices.** Because $T_\alpha$ is an $\alpha$-quantile, selective accuracy depends only on the ranking induced by $\hat{V}$. Monotone transformations of $\hat{V}$ leave selective accuracy exactly unchanged (Appendix G); the method is immune to any miscalibration that preserves ordering.

**Graceful degradation.** Under additive Gaussian noise on $\hat{V}$, performance degrades smoothly and gains over no abstention are retained at all noise levels tested, with Olympiad-Bench more sensitive than GSM8K (Appendix G).

### 6.5. Beyond Mathematical Reasoning: Toxicity Avoidance

The framework applies to any setting with a bounded reward. We test generalization to toxicity avoidance on RealToxicityPrompts (Gehman et al., 2020) with Qwen2.5-7B-Instruct. Figure 5 reports non-toxic response rate

among non-abstained samples. Dynamic value-thresholding reaches a perfect 1.00 rate at $\alpha = 0.3$ and dominates all baselines pointwise; the strongest baseline (self-assessment) only matches this from $\alpha = 0.6$. Full setup in Appendix L.

## 7. Discussion

Dynamic value-thresholding adds minimal overhead at inference: a single MLP forward pass per token, plus a one-time cost of generating trajectories and fitting the probe. This upfront cost is quickly amortized by the compute saved from early termination, which occurs in the first half of generation on average, even at low abstention rates (Section 6.1). Deployment is equally straightforward: the threshold $T_\alpha$ is the $\alpha$-quantile of estimated values on a small held-out set, and transfers to new data with mean absolute error below 1.2 percentage points (Section 6.4).

A natural concern is whether the method's reliance on value estimation makes it fragile. When the threshold is set to target a desired abstention rate $\alpha$, selective accuracy is exactly invariant to monotone reparametrizations of $\hat{V}$ and degrades gracefully under ranking-corrupting noise (Section 6.4). When instead $r_\perp$ is specified directly as the utility of the fallback mechanism, calibration becomes relevant; the miscalibration analysis in Appendix D suggests room for improvement in this regime.

The method is limited when hidden states do not encode sufficient information about eventual correctness, or when optimization fails to extract this signal. Our results suggest these issues are not severe in practice.

We highlight two directions for future work. First, our experiments use binary correctness or non-toxicity, but the theoretical results (Propositions 4.2–4.9) hold for any bounded reward. Substituting continuous rewards from a learned preference model and training the probe via MSE (Proposition B.3) would extend the method to tasks graded on a continuous scale. Second, our framework assumes a fixed fallback utility $r_\perp$, but in deployment, a question the model has nearly solved is worth continuing even at moderate confidence. Extending the framework to a state-dependent abstention reward $r_\perp(x, y_{1:t})$ would capture this.

## Acknowledgments

The authors thank Amy Mann, Yihong Chen, Xander Davies, and Sergio Calvo Ordoñez for proofreading and for suggestions on ordering, notation, and section structure. This work was supported by the EPSRC through the StatML CDT and by the Rhodes Trust. H. D. was supported by the Horizon Europe grant 101213369 DVPS. Part of this work was conducted during an internship at Amazon. P. R. was funded by UK Research and Innovation (UKRI) under the UK government's Horizon Europe funding guarantee [grant number EP/Y028333/1]. Research reported in this publication was supported by an Amazon Research Award, Fall 2024.

## Impact Statement

Dynamic value-thresholding reduces wasted computation by terminating unpromising reasoning traces early, which at scale could lower the energy consumption and environmental footprint of LLM inference. Beyond efficiency, principled abstention may improve reliability in high-stakes domains, such as medical reasoning, legal analysis, or financial decision-making, where confidently incorrect outputs cause disproportionate harm. By enabling models to defer appropriately to human experts or more capable systems, this work helps users develop calibrated trust in when to rely on model outputs. However, well-functioning abstention mechanisms could induce over-reliance: users may assume models will always abstain when uncertain. Finally, if abstention rates vary systematically across problem types or user populations, this could create disparities in service quality.

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

# A. Additional Related Works

Our work intersects with several research areas: abstention mechanisms for LLMs, selective classification theory, inference-time compute allocation, adaptive computation methods, and reward modeling for reasoning. We survey each area and position our contributions.

## A.1. Abstention in LLMs

Wen et al. (2024) provide a comprehensive survey of abstention in LLMs, organizing the literature along two axes. The first axis captures *why* abstention occurs, distinguishing three perspectives: the *query perspective* (is the question answerable at all?), the *model knowledge perspective* (does this model have the capability to answer correctly?), and the *human values perspective* (should the model answer, given safety and ethical considerations?). The second axis captures *when* the abstention decision is made during inference: *input-processing* methods decide before generation based on query properties, *in-processing* methods decide during generation based on model internals, and *output-processing* methods decide after complete generation.

The query perspective addresses whether a question is inherently answerable—for instance, ambiguous questions like "When did the war end?" (Min et al., 2020), questions requiring unavailable context (Rajpurkar et al., 2018), or questions beyond any knowledge such as future predictions (Amayuelas et al., 2023). The human values perspective addresses whether the model *should* answer, covering safety concerns like harmful queries (Wang et al., 2024), privacy violations, and toxic content generation (Gehman et al., 2020).

Our work focuses on the *model knowledge perspective* for reasoning tasks with verifiable answers. Here, questions are well-posed and safe to answer; the challenge is predicting whether *this particular model* will answer correctly. This perspective encompasses probing internal representations (Azaria & Mitchell, 2023), uncertainty estimation via token likelihoods or semantic entropy (Kuhn et al., 2023), and calibration-based methods (Jiang et al., 2021).

We note a terminological distinction from Wen et al.'s framework. Their "input-processing" refers to methods assessing query answerability or safety before generation. In contrast, our use of "input processing" in the main text refers to predicting *model success* from the prompt alone—a model-knowledge question evaluated at $t = 0$. Our baselines (constant step probing, self-assessment) fall into this category: they attempt to predict whether the model will succeed, not whether the question is inherently answerable.

Within the model knowledge perspective, existing in-processing methods estimate uncertainty via internal state probing or token likelihoods, then threshold these estimates to trigger abstention (Azaria & Mitchell, 2023; Kadavath et al., 2022). Recent work by Zhang et al. (2025) implements early exit by evaluating intermediate answer correctness at reasoning chunk boundaries using probing classifiers. Our approach differs in two respects. First, we threshold on the *value function*—the probability of *eventual* success conditioned on the current state—rather than intermediate correctness. A correct intermediate answer may later be revised incorrectly, and vice versa; what matters for abstention is whether the final answer will be correct. Second, we provide theoretical guarantees: we prove that value-thresholding dominates natural baselines (Propositions 4.2–4.4), characterize when it is optimal (Proposition 4.7), and bound the magnitude of improvement (Proposition 4.9). The RL formulation gives semantic meaning to the threshold $r_\perp$ as the utility of the fallback mechanism, providing practitioners direct control over the accuracy-efficiency tradeoff.

Abstention has also been studied in multi-turn agentic settings: Bonagiri et al. (2025) show that prompting tool-augmented agents to quit when uncertain substantially improves safety with minimal helpfulness loss, complementing our focus on within-generation abstention for reasoning tasks.

## A.2. Selective Classification

The theoretical foundations for abstention trace back to selective classification, also known as classification with a reject option. El-Yaniv & Wiener (2010) established the foundational framework by characterizing the *risk-coverage trade-off*: how much coverage must be sacrificed to achieve a target accuracy. They showed that for noise-free settings with finite hypothesis classes, a consistent selective strategy can achieve perfect learning with coverage approaching 1 as sample size grows.

Geifman & El-Yaniv (2017) extended these ideas to deep neural networks, proposing to use the softmax response (maximum softmax probability) as a confidence score for selective prediction. Given a trained neural network, they construct a selective

classifier that rejects instances to guarantee a desired risk level with high probability. Their method demonstrated that selective classification can enable DNNs to operate in mission-critical applications with formal accuracy guarantees.

Subsequent work has explored alternatives to softmax-based confidence, including SelectiveNet (Geifman & El-Yaniv, 2019), which trains a model to jointly optimize classification and rejection, ensemble-based approaches (Lakshminarayanan et al., 2017), and conformal methods that utilize a held-out calibration set to achieve the desired selective accuracy for any tunable black-box model (Angelopoulos et al., 2025). Our work differs from this literature by operating in the sequential generation setting, where the decision to abstain can be made at any token position rather than once per input.

### A.3. Inference-Time Compute Allocation

A key motivation for dynamic value-thresholding is efficient allocation of inference-time compute. Recent work has studied how to allocate computation adaptively across problems of varying difficulty. Best-of-N sampling generates multiple completions and selects the best according to a reward model. Snell et al. (2025) analyze optimal compute allocation between search breadth (number of samples) and depth (generation length). Li et al. (2024) propose early-stopping self-consistency, which reduces sampling costs by terminating when answer agreement is reached, cutting samples by over 30% on mathematical reasoning benchmarks. Speculative decoding (Leviathan et al., 2023) uses small models to draft tokens verified by larger models.

Manvi et al. (2024) introduce a generative self-evaluation scheme where the LLM predicts mid-generation the probability that restarting would yield a better response. This prediction requires only generating a single token and enables adaptive sample pruning without an external reward model—they show 50–75% of samples can be pruned early with minimal performance loss. Their work is closely related to ours: both leverage mid-generation signals to allocate compute efficiently. We differ in formulation (value-thresholding for abstention vs. restart probability for sample selection) and in providing theoretical guarantees for the stopping rule.

Our dynamic value-thresholding can be viewed as a compute allocation strategy: we allocate zero additional tokens to unpromising traces by terminating them early. Unlike best-of-N, which requires generating all samples before selection, our method makes incremental decisions, enabling real-time intervention. The threshold $r_\perp$ directly controls the accuracy-efficiency tradeoff: higher thresholds terminate more traces early, saving compute at the cost of fewer successful completions.

### A.4. Early Exit and Adaptive Computation

The principle of adaptive computation has been explored extensively in neural network architectures. Early exit methods allow computation to terminate at intermediate layers when confident (Teerapittayanon et al., 2016; Huang et al., 2018). For transformers, these methods typically add classifiers at intermediate layers and exit when confidence exceeds a threshold (Schuster et al., 2022; Elbayad et al., 2020).

Our approach applies this principle to the sequence dimension rather than the depth dimension: we exit early in the token sequence when the value function drops below the abstention threshold. This is complementary to layer-wise early exit—both could be combined for more efficient inference.

### A.5. LLM Calibration and Self-Knowledge

For dynamic value-thresholding to be effective, the model must have access to reliable signals about its likelihood of success. A foundational question is whether LLMs "know what they know." Kadavath et al. (2022) provided affirmative evidence: they showed that larger models are well-calibrated on multiple-choice and true/false questions when provided in appropriate formats. They introduced P(True), a self-evaluation probability where the model assesses its own generated answers, and P(IK), the probability that "I know" the answer without reference to a specific response. Both quantities showed promising calibration and scaling properties, though P(IK) struggled to generalize to new tasks.

Azaria & Mitchell (2023) demonstrated that LLM hidden states encode information about truthfulness. They trained classifiers on hidden layer activations to detect whether statements are true or false, achieving 71–83% accuracy depending on the base model. This finding—that internal representations contain veracity information beyond what the model expresses—motivates our use of hidden states for value function estimation.

### A.6. Uncertainty Estimation in Natural Language Generation

Beyond calibration of discrete judgments, uncertainty estimation in natural language generation faces unique challenges due to *semantic equivalence*: different sentences can express the same meaning. Kuhn et al. (2023) addressed this by introducing semantic entropy, which measures uncertainty over meanings rather than tokens. Their method clusters model samples by semantic equivalence and computes entropy in this meaning space. Semantic entropy is more predictive of model accuracy than token-level baselines on question-answering tasks.

Other approaches include self-consistency (Wang et al., 2023), which generates multiple responses and selects by majority vote, and linguistic calibration (Lin et al., 2022), which trains models to express uncertainty in natural language. Our value function approach can be viewed as estimating a specific form of confidence—the probability of eventual correctness—from hidden states, without requiring multiple samples or explicit verbalization.

### A.7. The Value of Mid-Generation Evaluation

A natural question is whether abstention decisions should be made from the prompt alone or whether observing partial generation provides meaningful additional signal. The evidence presents an interesting picture.

Afzal et al. (2025) find that probing classifiers predict Chain-of-Thought success surprisingly well even before a single token is generated, achieving 60–76% accuracy across datasets. This suggests that LLM representations encode substantial information about eventual correctness from the outset. However, they also observe that later reasoning steps do not *always* improve prediction—a finding we interpret as reflecting the heterogeneity of reasoning traces rather than the irrelevance of mid-generation information.

Complementary evidence suggests that generation reveals additional signal. Zhang et al. (2025) probe hidden states in reasoning models and find that predictive accuracy improves as generation progresses toward intermediate answers. Their early-exit strategy, which terminates generation when probe confidence exceeds a threshold, reduces inference tokens by 24% without accuracy loss—demonstrating that mid-generation signals enable efficiency gains unattainable at $t = 0$. Similarly, Manvi et al. (2024) show that LLMs can predict mid-generation whether restarting would yield a better response; this capability enables pruning 50–75% of samples early in generation with minimal performance degradation, further demonstrating that partial generations contain actionable information about eventual quality.

Qian et al. (2025) provide a theoretical lens on this phenomenon. Tracking mutual information between hidden representations and correct answers during generation, they observe that MI exhibits sudden peaks at "thinking tokens" (e.g., "Wait", "Therefore") corresponding to moments of reflection or logical transition. They prove that higher cumulative MI implies tighter bounds on prediction error. This non-monotonic information structure—where key insights crystallize at specific moments—explains why dynamic monitoring outperforms fixed-position evaluation.

Our framework formalizes this intuition. The value function $V^\pi(x, y_{1:t})$ captures all information relevant to predicting eventual success from state $(x, y_{1:t})$. By thresholding on this quantity throughout generation, we can exploit both early signals of likely failure and later moments when success becomes assured. Propositions 4.2–4.9 characterize exactly when and by how much this dynamic approach improves upon static alternatives.

## B. Theoretical Proofs

### B.1. Verification of the Necessary Condition

We verify that $a(\pi)$ satisfies Equation (7). Suppose first that $V_\beta(x, y_{1:t-1}; \pi) < r_\perp$. Then $a(\pi)$ abstains, so $y_t = \perp$ almost surely and $V_\beta(x, y_{1:t}; a(\pi)) = 0$. Thus

$$\mathbb{E}_{y_t \sim s(\pi^\dagger)(\cdot | x, y_{1:t-1})}\big[R_\beta(x, y_{1:t}; a(\pi)) + V_\beta(x, y_{1:t}; a(\pi))\big] = \mathbb{E}_{y_t \sim s(\pi^\dagger)(\cdot | x, y_{1:t-1})}\big[R_\beta(x, y_{1:t}; \pi)\big] \leq V_\beta(x, y_{1:t-1}; \pi) < r_\perp.$$

Conversely, if $V_\beta(x, y_{1:t-1}; \pi) \geq r_\perp$, then $a(\pi)$ follows $\pi$, and by Lemma 4.1:

$$\mathbb{E}_{y_t \sim s(\pi^\dagger)(\cdot | x, y_{1:t-1})}\big[R_\beta(x, y_{1:t}; a(\pi)) + V_\beta(x, y_{1:t}; a(\pi))\big]$$
$$= V_\beta(x, y_{1:t-1}; a(\pi))$$
$$\geq \max(V_\beta(x, y_{1:t-1}; \pi), r_\perp)$$
$$\geq r_\perp.$$

## B.2. Proof of Lemma 4.1

*Proof.* We prove by induction.

**Base Case I** ($t = T - 1$): At the final time step, the valid actions are $y^T \in \{\text{eos}, \perp\}$. The value of the augmented policy is:

$$V_\beta(x, y_{1:T-1}; a(\pi)) = a(\pi)(\perp \mid x, y_{1:T-1}) \cdot r_\perp \tag{14}$$

$$+ a(\pi)(\text{eos} \mid x, y_{1:T-1}) \left( r(x, [y_{1:T-1}, \text{eos}]) - \beta \log \frac{s(a(\pi))(\text{eos} \mid x, y_{1:T-1})}{\pi_{\text{ref}}(\text{eos} \mid x, y_{1:T-1})} \right). \tag{15}$$

We consider the two distinct cases defined by Eq. 8:

1. If $V_\beta(x, y_{1:T-1}; \pi) < r_\perp$: The policy abstains $(a(\pi)(\perp|x, y_{1:T-1}) = 1)$. Thus, $V_\beta(x, y_{1:T-1}; a(\pi)) = r_\perp > V_\beta(x, y_{1:T-1}; \pi)$.

2. If $V_\beta(x, y_{1:T-1}; \pi) \geq r_\perp$: The policy does not abstain $(a(\pi)(\perp|x, y_{1:T-1}) = 0)$. Consequently,

$$s(a(\pi))([y_{1:T-1}, \text{eos}] \mid x) = a(\pi)([y_{1:T-1}, \text{eos}] \mid x) = \pi([y_{1:T-1}, \text{eos}] \mid x).$$

   The value becomes

$$V_\beta(x, y_{1:T-1}; a(\pi)) = r(x, [y_{1:T-1}, \text{eos}]) - \beta \log \frac{\pi([y_{1:T-1}, \text{eos}] \mid x)}{\pi_{\text{ref}}([y_{1:T-1}, \text{eos}] \mid x)} = V_\beta(x, y_{1:T-1}; \pi) \geq r_\perp.$$

Combining these, $V_\beta(x, y_{1:T-1}; a(\pi)) = \max(r_\perp, V_\beta(x, y_{1:T-1}; \pi))$. There are no reachable terminal states.

**Base Case II** ($t = T - 2$): We again consider two cases:

1. If $V_\beta(x, y_{1:T-2}; \pi) < r_\perp$: The policy abstains $(a(\pi)(\perp|x, y_{1:T-2}) = 1)$. Thus, $V_\beta(x, y_{1:T-2}; a(\pi)) = r_\perp > V_\beta(x, y_{1:T-2}; \pi)$.

2. If $V_\beta(x, y_{1:T-2}; \pi) \geq r_\perp$: The policy does not abstain $(a(\pi)(\perp|x, y_{1:T-2}) = 0)$. Again, $s(a(\pi))([y_{1:T-2}, \text{eos}]|x) = a(\pi)([y_{1:T-2}, \text{eos}]|x) = \pi([y_{1:T-2}, \text{eos}]|x)$. The value becomes

$$V_\beta(x, y_{1:T-2}; a(\pi)) = \pi(\text{eos}|x, y_{1:T-2}) \left( r(x, [y_{1:T-2}, \text{eos}]) - \beta \log \frac{\pi([y_{1:T-2}, \text{eos}] \mid x)}{\pi_{\text{ref}}([y_{1:T-2}, \text{eos}] \mid x)} \right)$$

$$+ \sum_{y_{T-1} \in \mathcal{V}} \pi(y_{T-1} \mid x, y_{1:T-2}) V_\beta(x, [y_{1:T-2}, y_{T-1}]; a(\pi))$$

$$\geq \pi(\text{eos}|x, y_{1:T-2}) \left( r(x, [y_{1:T-2}, \text{eos}]) - \beta \log \frac{\pi([y_{1:T-2}, \text{eos}] \mid x)}{\pi_{\text{ref}}([y_{1:T-2}, \text{eos}] \mid x)} \right)$$

$$+ \sum_{y_{T-1} \in \mathcal{V}} \pi(y_{T-1} \mid x, y_{1:T-2}) V_\beta(x, [y_{1:T-2}, y_{T-1}]; \pi)$$

$$= V_\beta(x, y_{1:T-2}; \pi) \geq r_\perp.$$

Here, if for all $y_{T-1} \in \mathcal{V}$, $\pi(y_{T-1}|x, y_{1:T-2}) < 1$, the second inequality, stemming from Base Case I, holds strictly if and only if there is a reachable future state $x, [y_{1:T-2}, y_{T-1}]$ for which

$$V_\beta(x, [y_{1:T-2}, y_{T-1}]; a(\pi)) = r_\perp > V_\beta(x, [y_{1:T-2}, y_{T-1}]; \pi).$$

**Inductive Step:** Assume Eq. 9 holds for all states at step $t + 1$. We analyze the value at step $t$. Again, we split by the decision rule in Eq. 8:

*Case 1: Abstention ($V_\beta(x, y_{1:t}; \pi) < r_\perp$).* Here, $a(\pi)$ selects $\perp$ with probability 1. The sequence terminates, yielding:

$$V_\beta(x, y_{1:t}; a(\pi)) = r_\perp > V_\beta(x, y_{1:t}; \pi).$$

*Case 2: Continuation ($V_\beta(x, y_{1:t}; \pi) \geq r_\perp$).* Here, $a(\pi)$ selects tokens $y_t \in \mathcal{V}$ following $\pi(y_t \mid x, y_{1:t-1})$. As such

$$V_\beta(x, y_{1:t}; a(\pi)) = \pi(\text{eos}|x, y_{1:t}) \left( r(x, [y_{1:t}, \text{eos}]) - \beta \log \frac{\pi([y_{1:t}, \text{eos}] \mid x)}{\pi_{\text{ref}}([y_{1:t}, \text{eos}] \mid x)} \right)$$

$$+ \sum_{y_{t+1} \in \mathcal{V}} \pi(y_{t+1} \mid x, y_{1:t}) V_\beta(x, [y_{1:t}, y_{t+1}]; a(\pi))$$

$$\geq \pi(\text{eos}|x, y_{1:t}) \left( r(x, [y_{1:t}, \text{eos}]) - \beta \log \frac{\pi([y_{1:t}, \text{eos}] \mid x)}{\pi_{\text{ref}}([y_{1:t}, \text{eos}] \mid x)} \right)$$

$$+ \sum_{y_{t+1} \in \mathcal{V}} \pi(y_{t+1} \mid x, y_{1:t}) V_\beta(x, [y_{1:t}, y_{t+1}]; \pi)$$

$$= V_\beta(x, y_{1:t}; \pi) \geq r_\perp.$$

Again, if for all $y_{t+1} \in \mathcal{V}$, $\pi(y_{t+1}|x, y_{1:t}) < 1$, the second inequality, stemming from Induction hypothesis, holds strictly if and only if there is a reachable future state for which

$$V_\beta(x, y_{1:t+1}; a(\pi)) = r_\perp > V_\beta(x, y_{1:t+1}; \pi).$$

Thus, the induction is complete. □

### B.3. Proof of Proposition 4.4

*Proof of Proposition 4.4.* We work on the probability space of trajectories generated by the base policy: sample $x \sim \rho$ and $y \sim \pi(\cdot \mid x)$. Both $f(\pi; k)$ and $a(\pi)$ can be viewed as stopping rules applied to this common stochastic process: they follow $\pi$ for token generation but may terminate early by abstaining.

Define $R_f$ and $R_a$ as the rewards received by each policy:

$$R_f = \mathbb{I}\{V_{k-1} < r_\perp\} \cdot r_\perp + \mathbb{I}\{V_{k-1} \geq r_\perp\} \cdot R_\beta(x, y; \pi),$$
$$R_a = \mathbb{I}\{\tau < \infty\} \cdot r_\perp + \mathbb{I}\{\tau = \infty\} \cdot R_\beta(x, y; \pi),$$

where $V_t = V_\beta(x, y_{1:t}; \pi)$, $\tau = \min\{t \geq 1 : V_{t-1} < r_\perp\}$ with $\tau = \infty$ if no such $t$ exists, and $R_\beta(x, y; \pi)$ is the total reward defined in equation (2).

Since both policies follow $\pi$ when not abstaining, we have $J_\beta(f(\pi; k)) = \mathbb{E}[R_f]$ and $J_\beta(a(\pi)) = \mathbb{E}[R_a]$. We analyze these expectations by conditioning on the prefix $(x, y_{1:k-1})$, which determines both $\tau$ (restricted to $\{1, \ldots, k-1, \geq k\}$) and $V_{k-1}$.

**Expected reward under $f(\pi; k)$.**

The policy $f(\pi; k)$ follows $\pi$ until position $k$, then applies the abstention rule. Conditioned on $(x, y_{1:k-1})$, the future trajectory $y_{k:T}$ is random, drawn from $\pi(\cdot \mid x, y_{1:k-1})$. The conditional expected reward is:

- If $V_{k-1} < r_\perp$: $f(\pi; k)$ abstains at position $k$, so $R_f = r_\perp$ almost surely.

- If $V_{k-1} \geq r_\perp$: $f(\pi; k)$ continues with $\pi$, so $\mathbb{E}[R_f \mid x, y_{1:k-1}] = V_{k-1}$.

Thus:

$$\mathbb{E}[R_f \mid x, y_{1:k-1}] = \max(r_\perp, V_{k-1}). \tag{16}$$

**Expected reward under $a(\pi)$.**

The abstention time $\tau$ is determined by the prefix $(x, y_{1:k-1})$. Conditioned on $(x, y_{1:k-1})$:

- If $\tau < k$: the policy already abstained at position $\tau < k$, so $R_a = r_\perp$ almost surely.

- If $\tau \geq k$: the policy reached state $(x, y_{1:k-1})$ without abstaining. The future reward depends on subsequent abstention decisions and the terminal reward, yielding $\mathbb{E}[R_a \mid x, y_{1:k-1}] = V_\beta(x, y_{1:k-1}; a(\pi))$.

Thus:

$$\mathbb{E}[R_a \mid x, y_{1:k-1}] = \mathbb{I}\{\tau < k\} \cdot r_\perp + \mathbb{I}\{\tau \geq k\} \cdot V_\beta(x, y_{1:k-1}; a(\pi)). \tag{17}$$

**Computing the gap.**

Subtracting (17) from (16):

$$\mathbb{E}[R_f - R_a \mid x, y_{1:k-1}] = \mathbb{I}\{\tau < k\} \cdot (\max(r_\perp, V_{k-1}) - r_\perp)$$
$$+ \mathbb{I}\{\tau \geq k\} \cdot (\max(r_\perp, V_{k-1}) - V_\beta(x, y_{1:k-1}; a(\pi))). \tag{18}$$

We analyze each term.

*First term* $(\tau < k)$: The contribution is $\max(r_\perp, V_{k-1}) - r_\perp = (V_{k-1} - r_\perp)^+$.

*Second term* $(\tau \geq k)$: By definition of $\tau$, we have $V_s \geq r_\perp$ for all $s < k$. In particular, $V_{k-1} \geq r_\perp$, so $\max(r_\perp, V_{k-1}) = V_{k-1}$.

By Lemma 4.1:

$$V_\beta(x, y_{1:k-1}; a(\pi)) \geq \max(r_\perp, V_{k-1}) = V_{k-1}.$$

Therefore, the second term satisfies $V_{k-1} - V_\beta(x, y_{1:k-1}; a(\pi)) \leq 0$.

**Combining.**

From (18) and the analysis above:

$$\mathbb{E}[R_f - R_a \mid x, y_{1:k-1}] \leq \mathbb{I}\{\tau < k\} \cdot (V_{k-1} - r_\perp)^+.$$

By the tower property, taking expectations over $(x, y_{1:k-1})$ drawn from $x \sim \rho$ and $y_{1:k-1} \sim \pi(\cdot \mid x)$:

$$J_\beta(f(\pi; k)) - J_\beta(a(\pi)) = \mathbb{E}[R_f] - \mathbb{E}[R_a] = \mathbb{E}\left[\mathbb{E}[R_f - R_a \mid x, y_{1:k-1}]\right] \leq \mathbb{E}\left[\mathbb{I}\{\tau < k\} \cdot (V_{k-1} - r_\perp)^+\right].$$

Rearranging yields the bound (11).

**Characterizing strict inequality.**

From (18), the bound (11) is tight if and only if the second term equals zero almost surely. The second term is $\mathbb{I}\{\tau \geq k\} \cdot (V_{k-1} - V_\beta(x, y_{1:k-1}; a(\pi)))$, which is nonpositive. It equals zero at a state $(x, y_{1:k-1})$ with $\tau \geq k$ if and only if $V_\beta(x, y_{1:k-1}; a(\pi)) = V_{k-1}$.

By Lemma 4.1, $V_\beta(x, y_{1:k-1}; a(\pi)) > V_{k-1}$ if and only if there exists a state $(x, y_{1:t})$ with $t \geq k$ reachable from $(x, y_{1:k-1})$ under $\pi$ such that $V_t < r_\perp$.

Therefore, strict inequality in (11) holds if and only if there exists a reachable $(x, y_{1:k-1})$ with $\tau \geq k$ from which such a state is reachable. The condition $\tau \geq k$ is equivalent to $V_s \geq r_\perp$ for all $s < k$.

**When the correction term vanishes.**

The actual gap is $J_\beta(f(\pi; k)) - J_\beta(a(\pi)) = (\text{correction term}) + (\text{second term in (18)})$, where the second term is nonpositive. When the correction term equals zero, we have $J_\beta(f(\pi; k)) - J_\beta(a(\pi)) = (\text{second term}) \leq 0$, with strict inequality if and only if the second term is strictly negative somewhere. This is precisely the condition for strict inequality in (11). □

*Proof of Corollary 4.5.* If $\mathbb{P}(\tau < k) = 0$, then the correction term in (11) vanishes. The strict inequality characterization from Proposition 4.4 applies directly; note that condition (3) is automatically satisfied since $\tau \geq k$ for all reachable prefixes. □

*Proof of Corollary 4.6.* If the sub-level set is absorbing for $t < k - 1$, then conditioned on $\tau < k$, we have $V_{k-1} < r_\perp$ almost surely. Thus $(V_{k-1} - r_\perp)^+ = 0$ whenever $\tau < k$, and the correction term in (11) vanishes. □

### B.4. Proof of Proposition 4.7

*Proof.* Recall that for $\beta = 0$, the KL penalty term vanishes and the value function simplifies to the expected terminal reward:

$$V_0(x, y_{1:t}; \pi) = \mathbb{E}_{y \sim \pi(\cdot | x, y_{1:t})} \left[ r(x, y) \right]. \tag{19}$$

We prove two sub-claims for all reachable, non-terminal states $(x, y_{1:t})$:

1. For all $\pi_0^* \in \operatorname{argmax}_\pi J_0(\pi)$, we have $V_0(x, y_{1:t}; \pi_0^*) = \max_{y' \in \mathcal{V}^T : y'_{1:t} = y_{1:t}} r(x, y')$.

2. For all $\pi_0^* \in \operatorname{argmax}_\pi J_0(\pi)$, we have $V_0(x, y_{1:t}; a(\pi_0^*)) = \max(r_\perp, V_0(x, y_{1:t}; \pi_0^*))$.

Combining these sub-claims, for any $x$ in the support of $\rho$:

$$V_0(x, y_{1:0}; a(\pi_0^*)) = \max \left( r_\perp, V_0(x, y_{1:0}; \pi_0^*) \right) \tag{20}$$

$$= \max \left( r_\perp, \max_{y \in \mathcal{V}^T} r(x, y) \right) \tag{21}$$

$$= \max_{y \in \mathcal{V}^{\dagger T}} r(x, y), \tag{22}$$

where the last equality follows from the fact that any trajectory in $\mathcal{V}^{\dagger T}$ either terminates in $\perp$ (yielding reward $r_\perp$) or completes in $\mathcal{V}^T$ (yielding $r(x, y)$).

Taking expectations over $x \sim \rho$:

$$J_0(a(\pi_0^*)) = \mathbb{E}_{x \sim \rho} \left[ \max_{y \in \mathcal{V}^{\dagger T}} r(x, y) \right]. \tag{23}$$

We now show this is an upper bound for any policy $\pi^\dagger \in \Pi_{\mathcal{V}^\dagger}$. For any such policy:

$$J_0(\pi^\dagger) = \mathbb{E}_{x \sim \rho, y \sim \pi^\dagger(\cdot | x)} \left[ r^\dagger(x, y) \right] \tag{24}$$

where $r^\dagger(x, y) = r_\perp$ if $\perp \in y$ and $r^\dagger(x, y) = r(x, y)$ otherwise. For each $x$, the expectation over $y$ is a convex combination of rewards, each of which is at most $\max_{y \in \mathcal{V}^{\dagger T}} r(x, y)$. Therefore:

$$J_0(\pi^\dagger) \leq \mathbb{E}_{x \sim \rho} \left[ \max_{y \in \mathcal{V}^{\dagger T}} r(x, y) \right] = J_0(a(\pi_0^*)). \tag{25}$$

It remains to prove the sub-claims.

**Proof of Sub-claim 1.** Fix a reachable, non-terminal state $(x, y_{1:t})$. Let $v_{\max}^* = \max_{y' \in \mathcal{V}^T : y'_{1:t} = y_{1:t}} r(x, y')$ denote the maximum achievable reward consistent with the prefix $y_{1:t}$, and let $y^*$ be a completion attaining this maximum.

Suppose for contradiction that $V_0(x, y_{1:t}; \pi_0^*) < v_{\max}^*$. This implies $\pi_0^*$ assigns positive probability to completions with reward strictly less than $v_{\max}^*$.

Construct a modified policy $\pi'$ identical to $\pi_0^*$ except that, upon reaching state $(x, y_{1:t})$, it deterministically follows the optimal completion $y^*$:

$$\pi'(y_k \mid x, \tilde{y}_{1:k-1}) = \begin{cases} \mathbb{I}\{y_k = y_k^*\}, & \text{if } k > t \text{ and } \tilde{y}_{1:t} = y_{1:t} \\ \pi_0^*(y_k \mid x, \tilde{y}_{1:k-1}), & \text{otherwise.} \end{cases} \tag{26}$$

Under $\pi'$, the value at $(x, y_{1:t})$ is $V_0(x, y_{1:t}; \pi') = r(x, y^*) = v_{\max}^*$.

Since $\pi'$ differs from $\pi_0^*$ only for trajectories passing through $(x, y_{1:t})$:

$$J_0(\pi') - J_0(\pi_0^*) = \mathbb{E}_{x' \sim \rho} \left[ V_0(x', y_{1:0}; \pi') - V_0(x', y_{1:0}; \pi_0^*) \right] \tag{27}$$

$$= \rho(x) \cdot \pi_0^*(y_{1:t} \mid x) \cdot \left( V_0(x, y_{1:t}; \pi') - V_0(x, y_{1:t}; \pi_0^*) \right) \tag{28}$$

$$> 0, \tag{29}$$

where the final inequality follows from reachability ($\rho(x) \cdot \pi_0^*(y_{1:t} \mid x) > 0$) and $V_0(x, y_{1:t}; \pi') > V_0(x, y_{1:t}; \pi_0^*)$.

This contradicts the optimality of $\pi_0^*$. Therefore $V_0(x, y_{1:t}; \pi_0^*) = v_{\max}^*$.

**Proof of Sub-claim 2.** We proceed by backward induction on $t$.

*Base case ($t = T - 1$):* At the final non-terminal position, the only available action in $\mathcal{V}$ is eos, so $V_0(x, y_{1:T-1}; \pi_0^*) = r(x, y_{1:T-1} \circ \text{eos})$.

If $V_0(x, y_{1:T-1}; \pi_0^*) < r_\perp$, then $a(\pi_0^*)$ abstains, yielding:

$$V_0(x, y_{1:T-1}; a(\pi_0^*)) = r_\perp = \max(r_\perp, V_0(x, y_{1:T-1}; \pi_0^*)). \tag{30}$$

If $V_0(x, y_{1:T-1}; \pi_0^*) \geq r_\perp$, then $a(\pi_0^*)$ outputs eos, yielding:

$$V_0(x, y_{1:T-1}; a(\pi_0^*)) = V_0(x, y_{1:T-1}; \pi_0^*) = \max(r_\perp, V_0(x, y_{1:T-1}; \pi_0^*)). \tag{31}$$

*Inductive step:* Assume the claim holds for all reachable, non-terminal states at positions $t + 1, \ldots, T - 1$. Consider a reachable, non-terminal state $(x, y_{1:t})$.

*Case 1: $V_0(x, y_{1:t}; \pi_0^*) < r_\perp$.* By definition of $a(\pi)$ in Eq. 8, $a(\pi_0^*)$ abstains:

$$V_0(x, y_{1:t}; a(\pi_0^*)) = r_\perp = \max(r_\perp, V_0(x, y_{1:t}; \pi_0^*)). \tag{32}$$

*Case 2: $V_0(x, y_{1:t}; \pi_0^*) \geq r_\perp$.* By definition of $a(\pi)$, $a(\pi_0^*)$ follows $\pi_0^*$:

$$V_0(x, y_{1:t}; a(\pi_0^*)) = \mathbb{E}_{y_{t+1} \sim \pi_0^*(\cdot|x,y_{1:t})} \left[ R_0(x, y_{1:t+1}; a(\pi_0^*)) + V_0(x, y_{1:t+1}; a(\pi_0^*)) \right]. \tag{33}$$

Since $\beta = 0$, we have $R_0(x, y_{1:t+1}; a(\pi_0^*)) = R_0(x, y_{1:t+1}; \pi_0^*)$ for non-abstaining trajectories. Applying the inductive hypothesis to each reachable successor state:

$$V_0(x, y_{1:t}; a(\pi_0^*)) = \mathbb{E}_{y_{t+1} \sim \pi_0^*(\cdot|x,y_{1:t})} \left[ R_0(x, y_{1:t+1}; \pi_0^*) + \max(r_\perp, V_0(x, y_{1:t+1}; \pi_0^*)) \right]. \tag{34}$$

By Sub-claim 1, $V_0(x, y_{1:t}; \pi_0^*)$ equals the maximum reward achievable from $(x, y_{1:t})$, and this maximum is attained by some completion. Since $\pi_0^*$ is optimal, it places probability only on reward-maximizing continuations, so $V_0(x, y_{1:t+1}; \pi_0^*) = V_0(x, y_{1:t}; \pi_0^*)$ for all $y_{t+1}$ in the support of $\pi_0^*(\cdot \mid x, y_{1:t})$.

Since we are in Case 2, $V_0(x, y_{1:t}; \pi_0^*) \geq r_\perp$, so $\max(r_\perp, V_0(x, y_{1:t+1}; \pi_0^*)) = V_0(x, y_{1:t+1}; \pi_0^*)$ for these successor states. Therefore:

$$V_0(x, y_{1:t}; a(\pi_0^*)) = \mathbb{E}_{y_{t+1} \sim \pi_0^*(\cdot|x,y_{1:t})} \left[ R_0(x, y_{1:t+1}; \pi_0^*) + V_0(x, y_{1:t+1}; \pi_0^*) \right] \tag{35}$$
$$= V_0(x, y_{1:t}; \pi_0^*) \tag{36}$$
$$= \max(r_\perp, V_0(x, y_{1:t}; \pi_0^*)). \tag{37}$$

This completes the induction. $\qquad\square$

## B.5. Proof of Proposition 4.9

*Proof.* We work throughout under the assumptions $\beta = 0$ and $r(x, y) \in \{0, 1\}$, which imply $V_0(x, y_{1:t}; \pi) \in [0, 1]$ for all states.

### Step 1: Setup and stopping times.

Define the following stopping times on the probability space of trajectories generated by sampling $x \sim \rho$ and $y \sim \pi(\cdot \mid x)$:

- $\tau = \min\{t \geq 1 : V_0(x, y_{1:t-1}; \pi) < r_\perp\}$, the first timestep at which $a(\pi)$ abstains, with $\tau = \infty$ if no such $t$ exists.

- $c = \min\{t : y_t = \text{eos}\}$, the completion time under $\pi$.

Note that $\tau$ is determined by the trajectory $(x, y)$: it is the first position $t$ such that the value at the *preceding* state $(x, y_{1:t-1})$ falls below $r_\perp$.

### Step 2: Expressing the objective difference via trajectories from $\pi$.

The key observation is that both $J_0(a(\pi))$ and $J_0(\pi)$ can be expressed as expectations over trajectories generated by $\pi$. For $J_0(\pi)$, this is immediate:

$$J_0(\pi) = \mathbb{E}_{x \sim \rho} \mathbb{E}_{y \sim \pi(\cdot|x)}[r(x, y)].$$

For $J_0(a(\pi))$, we analyze what reward $a(\pi)$ obtains on a trajectory $y$ generated by $\pi$:

- If $\tau \leq c$: The policy $a(\pi)$ would abstain at step $\tau$ (before or at completion), receiving reward $r_\perp$.

- If $\tau > c$: The policy $a(\pi)$ follows $\pi$ until completion at step $c$, receiving reward $r(x, y)$.

Therefore:

$$J_0(a(\pi)) = \mathbb{E}_{x \sim \rho} \mathbb{E}_{y \sim \pi(\cdot|x)} \left[ r_\perp \cdot \mathbb{I}\{\tau \leq c\} + r(x, y) \cdot \mathbb{I}\{\tau > c\} \right].$$

Taking the difference:

$$
\begin{aligned}
J_0(a(\pi)) - J_0(\pi) &= \mathbb{E}_{x,y} \left[ r_\perp \cdot \mathbb{I}\{\tau \leq c\} + r(x, y) \cdot \mathbb{I}\{\tau > c\} - r(x, y) \right] \\
&= \mathbb{E}_{x,y} \left[ r_\perp \cdot \mathbb{I}\{\tau \leq c\} - r(x, y) \cdot \mathbb{I}\{\tau \leq c\} \right] \\
&= \mathbb{E}_{x,y} \left[ (r_\perp - r(x, y)) \cdot \mathbb{I}\{\tau \leq c\} \right].
\end{aligned}
\tag{38}
$$

### Step 3: Replacing $r(x, y)$ with the value function at abstention time.

We now show that (38) can be rewritten in terms of the value function at abstention time. The key is the following claim.

*Claim:* $\mathbb{E}_{x,y} \left[ r(x, y) \cdot \mathbb{I}\{\tau \leq c\} \right] = \mathbb{E}_{x,y} \left[ V_0(x, y_{1:\tau-1}; \pi) \cdot \mathbb{I}\{\tau \leq c\} \right].$

*Proof of claim.* We use the law of iterated expectations. Define the $\sigma$-algebra $\mathcal{G} = \sigma(x, y_{1:\tau-1})$ generated by the prompt and the trajectory up to (but not including) the abstention step. We will show that $\mathbb{I}\{\tau \leq c\}$ is $\mathcal{G}$-measurable, which allows us to apply the tower property.

To see that $\mathbb{I}\{\tau \leq c\}$ is $\mathcal{G}$-measurable, observe:

- The stopping time $\tau = \min\{t \geq 1 : V_0(x, y_{1:t-1}; \pi) < r_\perp\}$ is determined by $(x, y_{1:\tau-1})$, since $\tau = k$ iff $V_0(x, y_{1:t-1}; \pi) \geq r_\perp$ for $t < k$ and $V_0(x, y_{1:k-1}; \pi) < r_\perp$.

- The event $\{\tau \leq c\}$ is equivalent to $\{y_t \neq \texttt{eos}$ for all $t < \tau\}$, which depends only on $y_{1:\tau-1}$.

Thus $\mathbb{I}\{\tau \leq c\}$ is $\mathcal{G}$-measurable.

Now, by the tower property:

$$
\begin{aligned}
\mathbb{E}\left[ r(x, y) \cdot \mathbb{I}\{\tau \leq c\} \right] &= \mathbb{E}\left[ \mathbb{E}\left[ r(x, y) \cdot \mathbb{I}\{\tau \leq c\} \mid \mathcal{G} \right] \right] \\
&= \mathbb{E}\left[ \mathbb{I}\{\tau \leq c\} \cdot \mathbb{E}\left[ r(x, y) \mid \mathcal{G} \right] \right] \quad \text{(since } \mathbb{I}\{\tau \leq c\} \text{ is } \mathcal{G}\text{-measurable)} \\
&= \mathbb{E}\left[ \mathbb{I}\{\tau \leq c\} \cdot \mathbb{E}\left[ r(x, y) \mid x, y_{1:\tau-1} \right] \right] \\
&= \mathbb{E}\left[ \mathbb{I}\{\tau \leq c\} \cdot V_0(x, y_{1:\tau-1}; \pi) \right],
\end{aligned}
$$

where the last equality uses $V_0(x, y_{1:t}; \pi) = \mathbb{E}_{y \sim \pi(\cdot|x, y_{1:t})}[r(x, y)]$ for non-terminal states, which holds since $\beta = 0$. $\qquad \square$ (claim)

Substituting into (38):

$$J_0(a(\pi)) - J_0(\pi) = \mathbb{E}_{x,y} \left[ (r_\perp - V_0(x, y_{1:\tau-1}; \pi)) \cdot \mathbb{I}\{\tau \leq c\} \right].
\tag{39}$$

**Step 4: Bounding the value gap.**

We bound $r_\perp - V_0(x, y_{1:\tau-1}; \pi)$ by considering two cases.

*Case $\tau \geq 2$:* By definition of $\tau$:

- $V_0(x, y_{1:\tau-2}; \pi) \geq r_\perp$ (otherwise $\tau$ would be at most $\tau - 1$), and

- $V_0(x, y_{1:\tau-1}; \pi) < r_\perp$ (the condition triggering abstention at time $\tau$).

Combining these inequalities:
$$r_\perp - V_0(x, y_{1:\tau-1}; \pi) < r_\perp \leq V_0(x, y_{1:\tau-2}; \pi).$$

Rearranging:
$$r_\perp - V_0(x, y_{1:\tau-1}; \pi) < V_0(x, y_{1:\tau-2}; \pi) - V_0(x, y_{1:\tau-1}; \pi) + (r_\perp - r_\perp) = V_0(x, y_{1:\tau-2}; \pi) - V_0(x, y_{1:\tau-1}; \pi).$$

By the $L$-Lipschitz condition (Definition 4.8), which applies since $\tau \geq 2$ implies the state $(x, y_{1:\tau-1})$ has $\tau - 1 \geq 1$:
$$r_\perp - V_0(x, y_{1:\tau-1}; \pi) \leq |V_0(x, y_{1:\tau-2}; \pi) - V_0(x, y_{1:\tau-1}; \pi)| \leq L.$$

*Case $\tau = 1$:* Abstention occurs at the first step, meaning $V_0(x, y_{1:0}; \pi) < r_\perp$. The Lipschitz condition cannot be applied because there is no preceding state $y_{1:-1}$. However, since $\beta = 0$ and $r(x, y) \in \{0, 1\}$, the value function is a conditional probability:
$$V_0(x, y_{1:0}; \pi) = \mathbb{P}_\pi(r(x, y) = 1 \mid x) \in [0, 1].$$

In particular, $V_0(x, y_{1:0}; \pi) \geq 0$. Combined with $r_\perp \leq 1$, we obtain:
$$r_\perp - V_0(x, y_{1:0}; \pi) \leq r_\perp - 0 = r_\perp.$$

**Step 5: Assembling the bound.**

Define $\alpha_1 = \mathbb{P}(\tau = 1)$ and $\alpha_{\geq 2} = \mathbb{P}(\tau \geq 2, \tau \leq c)$. Note that $\tau = 1$ automatically implies $\tau \leq c$ (since $c \geq 1$ by definition), so $\alpha = \alpha_1 + \alpha_{\geq 2}$ is the total abstention rate.

From (39):
$$\begin{aligned}
J_0(a(\pi)) - J_0(\pi) &= \mathbb{E}\left[(r_\perp - V_0(x, y_{1:\tau-1}; \pi)) \cdot \mathbb{I}\{\tau \leq c\}\right] \\
&= \mathbb{E}\left[(r_\perp - V_0(x, y_{1:0}; \pi)) \cdot \mathbb{I}\{\tau = 1\}\right] + \mathbb{E}\left[(r_\perp - V_0(x, y_{1:\tau-1}; \pi)) \cdot \mathbb{I}\{\tau \geq 2, \tau \leq c\}\right].
\end{aligned}$$

Applying the bounds from Step 4 to each term:
$$\begin{aligned}
J_0(a(\pi)) - J_0(\pi) &\leq \mathbb{E}\left[r_\perp \cdot \mathbb{I}\{\tau = 1\}\right] + \mathbb{E}\left[L \cdot \mathbb{I}\{\tau \geq 2, \tau \leq c\}\right] \\
&= \alpha_1 \cdot r_\perp + \alpha_{\geq 2} \cdot L \\
&= \alpha_1 \cdot r_\perp + (\alpha - \alpha_1) \cdot L.
\end{aligned}$$

This completes the proof. □

## B.6. Value as Conditional Expectation

We first establish that the value function equals a conditional expectation of the terminal reward, then specialize to the $\beta = 0$ case with binary rewards used in the main text.

**Proposition B.1** (Value as Conditional Expectation). *For any $\beta \geq 0$ and any state $(x, y_{1:t})$:*
$$V_\beta(x, y_{1:t}; \pi) = \mathbb{E}_{y \sim \pi(\cdot | x, y_{1:t})}\left[R_\beta(x, y) \cdot \mathbb{I}\{t < c\}\right], \tag{40}$$

*where $c = \min\{k : y_k = \texttt{eos}\}$ is the completion time and $R_\beta(x, y) = r(x, y) - \beta \log \frac{\pi(y|x)}{\pi_{\text{ref}}(y|x)}$ is the terminal reward.*

*Proof.* We consider terminal and non-terminal states separately.

**Terminal states** ($t \geq c$): The sequence has already terminated, so no future rewards are collected. By the definition of the value function, $V_\beta(x, y_{1:t}; \pi) = 0$. The indicator $\mathbb{I}\{t < c\} = 0$ enforces this: the right-hand side equals $\mathbb{E}[R_\beta(x, y) \cdot 0] = 0$.

**Non-terminal states** ($t < c$): The indicator $\mathbb{I}\{t < c\} = 1$ almost surely for any completion $y \sim \pi(\cdot|x, y_{1:t})$, since the sequence has not yet terminated. The sparse reward structure (Equation 1) implies that all intermediate rewards are zero, with the entire reward concentrated at the terminal token. Thus:

$$V_\beta(x, y_{1:t}; \pi) = \mathbb{E}_{y \sim \pi(\cdot|x, y_{1:t})} \left[ \sum_{k=t+1}^{T} R_\beta(x, y_{1:k}; \pi) \right]$$
$$= \mathbb{E}_{y \sim \pi(\cdot|x, y_{1:t})} [R_\beta(x, y)]$$
$$= \mathbb{E}_{y \sim \pi(\cdot|x, y_{1:t})} [R_\beta(x, y) \cdot \mathbb{I}\{t < c\}],$$

where the last equality uses $\mathbb{I}\{t < c\} = 1$ almost surely. $\qquad\square$

**Specialization to $\beta = 0$ with binary rewards.** For $\beta = 0$, the terminal reward reduces to $R_0(x, y) = r(x, y)$. When $r(x, y) \in \{0, 1\}$ indicates correctness, Proposition B.1 implies that for non-terminal states ($t < c$):

$$V_0(x, y_{1:t}; \pi) = \mathbb{E}[r(x, y) \mid x, y_{1:t}] = \mathbb{P}_\pi(r(x, y) = 1 \mid x, y_{1:t}),$$

which is Equation 12 in the main text. For terminal states ($t \geq c$), $V_0(x, y_{1:t}; \pi) = 0$.

### B.7. BCE Recovers Value Function

**Proposition B.2** (BCE Recovers Value Function). *For $\beta = 0$ with binary rewards $r(x, y) \in \{0, 1\}$, let $\mathcal{H} = \{\hat{V}_t(\cdot; \theta) : \theta \in \Theta\}$ denote the hypothesis class, where $\hat{V}_t(x, y_{1:t}; \theta) = 0$ for terminal states by definition. Assume realizability: $V_0(\cdot, \cdot; \pi) \in \mathcal{H}$. Then the minimizer $\hat{V}_t^*$ of $\mathcal{L}(\theta)$ satisfies $\hat{V}_t^*(x, y_{1:t}) = V_0(x, y_{1:t}; \pi)$ for all states.*

*Proof.* Since $\mathcal{D}$ is generated by $x \sim \rho$ and $y \sim \pi(\cdot|x)$, the distribution over prefixes $(x, y_{1:t})$ induced by $\mathcal{D}$ matches the distribution of states visited by $\pi$ under $\rho$. We verify that $\hat{V}_t^* = V_0(x, y_{1:t}; \pi)$ for both terminal and non-terminal states.

**Terminal states** ($t \geq c$): We have $\hat{V}_t^* = 0$ by the architectural definition. From Appendix B.6, $V_0(x, y_{1:t}; \pi) = 0$ for terminal states.

**Non-terminal states** ($t < c$): The loss (Equation 13) sums only over non-terminal positions. The expected loss conditional on $(x, y_{1:t})$ is the standard cross-entropy:

$$\mathbb{E} \left[ -r(x, y) \log \hat{V}_t - (1 - r(x, y)) \log(1 - \hat{V}_t) \mid x, y_{1:t} \right].$$

Taking the derivative with respect to $\hat{V}_t$ and setting to zero:

$$\mathbb{E} \left[ -\frac{r(x, y)}{\hat{V}_t} + \frac{1 - r(x, y)}{1 - \hat{V}_t} \mid x, y_{1:t} \right] = 0.$$

Let $p = \mathbb{E}[r(x, y) \mid x, y_{1:t}]$ denote the conditional probability of correctness. Rearranging:

$$\frac{1 - p}{1 - \hat{V}_t} = \frac{p}{\hat{V}_t} \quad \Rightarrow \quad \hat{V}_t^* = p = \mathbb{P}_\pi(r(x, y) = 1 \mid x, y_{1:t}) = V_0(x, y_{1:t}; \pi),$$

where the final equality follows from Equation 12. $\qquad\square$

### B.8. MSE Recovers Value Function

**Proposition B.3** (MSE Recovers Value Function). *For any $\beta \geq 0$, let $\mathcal{D} = \{(x_i, y_i)\}_{i=1}^{N}$ be a dataset of trajectories where $x_i \sim \rho$ and $y_i \sim \pi(\cdot|x_i)$. Consider the mean squared error objective:*

$$\mathcal{L}_{\text{MSE}}(\theta) = \mathbb{E}_{(x,y) \sim \mathcal{D}} \left[ \sum_{t=0}^{c-1} \left( \hat{V}_t(x, y_{1:t}; \theta) - R_\beta(x, y) \right)^2 \right],$$

*where* $R_\beta(x, y) = r(x, y) - \beta \log \frac{\pi(y|x)}{\pi_{\text{ref}}(y|x)}$. *Let* $\hat{V}_t^*$ *denote the minimizer. Then* $\hat{V}_t^*(x, y_{1:t}) = V_\beta(x, y_{1:t}; \pi)$ *for all non-terminal states.*

*Proof.* Since $\mathcal{D}$ is generated by $x \sim \rho$ and $y \sim \pi(\cdot|x)$, the distribution over prefixes $(x, y_{1:t})$ induced by $\mathcal{D}$ matches the distribution of states visited by $\pi$ under $\rho$. The expected loss at position $t < c$ conditional on $(x, y_{1:t})$ is:

$$\mathbb{E}\left[\left(\hat{V}_t - R_\beta(x, y)\right)^2 \mid x, y_{1:t}\right].$$

Taking the derivative with respect to $\hat{V}_t$ and setting to zero:

$$2\,\mathbb{E}\left[\hat{V}_t - R_\beta(x, y) \mid x, y_{1:t}\right] = 0.$$

Rearranging:

$$\hat{V}_t^* = \mathbb{E}\left[R_\beta(x, y) \mid x, y_{1:t}\right] = V_\beta(x, y_{1:t}; \pi),$$

where the final equality follows from Proposition B.1. $\qquad\square$

## C. Algorithm Pseudocode

---

**Algorithm 1** Training the Value Function Estimator

---

**Require:** Policy $\pi$, dataset $\mathcal{D} = \{(x_i, y_i)\}_{i=1}^N$ with $x_i \sim \rho$, $y_i \sim \pi(\cdot \mid x_i)$, reward function $r : \mathcal{X} \times \mathcal{V}^* \to \{0, 1\}$
**Ensure:** Trained estimator $\hat{V}_\theta$ approximating $V_0(x, y_{1:t}; \pi)$
 1: Initialize parameters $\theta$
 2: **for** each epoch **do**
 3:     **for** each $(x, y) \in \mathcal{D}$ **do**
 4:        $c \leftarrow \min\{t : y_t = \texttt{eos}\}$ {Completion time}
 5:        **for** $t = 0, \ldots, c - 1$ **do**
 6:           Extract hidden state $h_t$ from $\pi$ at state $(x, y_{1:t})$
 7:           $\hat{V}_t \leftarrow \text{MLP}_\theta(h_t)$ {Estimate $V_0(x, y_{1:t}; \pi)$}
 8:           $\mathcal{L} \leftarrow \mathcal{L} + \ell(\hat{V}_t, r(x, y))$ {Eq. 13}
 9:        **end for**
10:     **end for**
11:     Update $\theta$ via gradient descent on $\mathcal{L}$
12: **end for**
13: **return** $\hat{V}_\theta$

---

---

**Algorithm 2** Dynamic Value-Thresholding at Inference

---

**Require:** Policy $\pi$, trained estimator $\hat{V}_\theta$, abstention reward $r_\perp \in (0, 1]$, prompt $x$, max length $T$
**Ensure:** Response $y \in \mathcal{V}^*$ or abstention $\perp$
  1: $t \leftarrow 0$
  2: **while** $t < T$ **do**
  3:      Run forward pass of $\pi$ on $(x, y_{1:t})$; let $h_t$ be the hidden state
  4:      $\hat{V}_t \leftarrow \text{MLP}_\theta(h_t)$ {Estimate $V_0(x, y_{1:t}; \pi)$}
  5:      **if** $\hat{V}_t < r_\perp$ **then**
  6:          **return** $\perp$ {Abstain per Eq. 8}
  7:      **end if**
  8:      Sample $y_{t+1} \sim \pi(\cdot \mid x, y_{1:t})$
  9:      **if** $y_{t+1} = \texttt{eos}$ **then**
10:          **return** $y_{1:t+1}$
11:      **end if**
12:      $t \leftarrow t + 1$
13: **end while**
14: **return** $y_{1:T}$ {Max length reached}

---

# D. Reward Objective Evaluation Methodology

Section 6.2 evaluates methods on the reward objective $J_\beta = \alpha \cdot r_\perp + (1 - \alpha) \cdot S$. Here we detail the methodology and the calibration analysis motivating it.

### D.1. The Evaluation Challenge: Relating Thresholds to $r_\perp$

The theoretical framework specifies $r_\perp$ as a threshold on the true value function: the optimal policy abstains when $V_\beta(x, y_{1:t}; \pi) < r_\perp$. Since $r_\perp$ operates on true values, it has a direct interpretation: it is the minimum expected future reward at which continuing generation is preferred to the fallback.

In practice, we do not have access to $V_\beta$. Instead, we threshold an estimate: abstain when $\hat{V}_t(x, y_{1:t}) < T_\alpha$, where $T_\alpha$ is chosen to achieve abstention rate $\alpha$. Because $\hat{V}_t \neq V_\beta$ in general, the threshold $T_\alpha$ on the estimate does not directly correspond to a threshold $r_\perp$ on the true value.

For dynamic abstention, let $\tau = \min\{t : \hat{V}_t < T_\alpha\}$ denote the abstention time: the first position at which the estimated value falls below the threshold. The value at abstention time, $\hat{V}_\tau$, is the natural quantity to calibrate: it represents the model's estimated expected reward at the moment the abstention decision is triggered. Crucially, by construction $\hat{V}_\tau < T_\alpha$ and $\hat{V}_\tau \approx T_\alpha$ (since abstention occurs at the first crossing). This tight relationship between the abstention-time value and the threshold simplifies the calibration problem relative to alternative formulations.

The relationship between $T_\alpha$ and the effective $r_\perp$ depends on the calibration quality of $\hat{V}_\tau$. If $\hat{V}_\tau$ is well-calibrated—meaning $\hat{V}_\tau \approx \mathbb{P}(\text{correct} \mid x, y_{1:\tau})$—then $T_\alpha$ directly estimates the effective $r_\perp$. When calibration is imperfect, methods with identical selective accuracy but different calibration properties will appear to achieve different rewards.

### D.2. Calibration Analysis

For dynamic abstention, the relevant quantity for calibration is the value at abstention time $\hat{V}_\tau$, not the pointwise estimates themselves. Since abstention occurs when $\hat{V}_\tau$ first drops below the threshold $T_\alpha$, we have $\hat{V}_\tau \approx T_\alpha$ by construction. This tight coupling means that the calibration of abstention-time values directly determines how well $T_\alpha$ estimates the effective $r_\perp$.

Figure 6 compares the calibration of the baseline method (value at $t = 0$) against the abstention-time values for our method. Both the prompt-based value estimate $\hat{V}_0$ and the abstention-time value $\hat{V}_\tau$ show some miscalibration across all settings.

## D.3. Recalibration Methodology

To enable fair comparison across methods with different calibration properties, we estimate the effective $r_\perp$ corresponding to each method's threshold $T_\alpha$. Recall that for $\beta = 0$ with binary rewards, the true value function equals the probability of correctness: $V_0(x, y_{1:t}; \pi) = \mathbb{P}(r(x, y) = 1 \mid x, y_{1:t})$. The effective $r_\perp$ at threshold $T_\alpha$ is therefore the true probability of correctness at the point where abstention is triggered.

We seek a transformation $g$ from estimated values to probabilities. Two properties are essential:

1. **Calibration**: For any probability $p$, among samples with $g(\hat{V}) \approx p$, the empirical frequency of correctness should be approximately $p$. This ensures that $g(T_\alpha)$ correctly estimates the true probability of correctness at the point where abstention is triggered.

2. **Monotonicity**: $g$ should be non-decreasing. This ensures coherence: higher estimated values correspond to higher true probabilities.

Isotonic regression provides a transformation satisfying both properties. For baseline methods that make decisions at $t = 0$, we fit isotonic regression on $(\hat{V}_0, \text{correctness})$ pairs across all samples. For dynamic abstention, we fit isotonic regression on $(\hat{V}_\tau, \text{correctness})$ pairs, where $\hat{V}_\tau$ is the value at the abstention time for samples that abstain at threshold $T_\alpha$. Since $\hat{V}_\tau \approx T_\alpha$ by construction, this calibration is performed separately for each threshold level to capture any threshold-dependent calibration effects. Let $g_{M,\alpha}$ denote the calibration function for method $M$ at threshold $T_\alpha$. We fit:

$$g_{M,\alpha} = \underset{g \in \mathcal{G}_\uparrow}{\arg\min} \sum_{i=1}^{N_\alpha} (y_i - g(\hat{V}_i))^2, \tag{41}$$

where $\mathcal{G}_\uparrow$ is the class of monotonically non-decreasing functions and the sum is over samples relevant to threshold $T_\alpha$.

For a chosen abstention rate $\alpha$, we estimate the reward objective as:

$$\hat{J}_{\text{calib}}(\alpha) = (1 - \alpha) \cdot \hat{S}(\alpha) + \alpha \cdot g_{M,\alpha}(T_\alpha), \tag{42}$$

where $\hat{S}(\alpha)$ is the empirical selective accuracy at rate $\alpha$, and $g_{M,\alpha}(T_\alpha)$ estimates the effective $r_\perp$.

**Why the $\hat{r}_\perp$ range is limited in Figure 3.** The x-axis in Figure 3 does not span $[0, 1]$ for several reasons:

1. **We test a finite set of abstention rates.** We evaluate at $\alpha \in \{0.02, 0.03, \ldots, 0.98\}$, not the full interval $[0, 1]$.

2. **Each abstention rate determines a threshold.** For each $\alpha$, we find the threshold $T_\alpha$ such that exactly fraction $\alpha$ of samples abstain.

3. **For each threshold, we fit isotonic regression separately.** Per the methodology above, calibration is performed independently for each threshold on samples that abstain at that threshold. This means different thresholds yield different calibration functions $g_{M,\alpha}$. Since each calibration is fit independently, there is no monotonicity guarantee across thresholds—the resulting $\hat{r}_\perp$ values reflect the empirical relationship between values and correctness for each threshold's specific set of abstaining samples.

4. **The range of $\hat{r}_\perp$ is determined by empirical accuracies at abstention boundaries.** For each threshold $T_\alpha$, the calibrated value $\hat{r}_\perp = g_{M,\alpha}(T_\alpha)$ estimates the true probability of correctness for samples near the abstention boundary. This is fundamentally bounded by the empirical accuracy of these samples:

   - *Lower bound:* At low abstention rates (e.g., $\alpha = 0.02$), only the lowest-value samples abstain. Even these worst-performing samples may have non-zero accuracy, so $\hat{r}_\perp > 0$.
   - *Upper bound:* At high abstention rates (e.g., $\alpha = 0.98$), samples near the threshold have high value estimates but are not perfectly accurate. Thus $\hat{r}_\perp < 1$ for thresholds where we have sufficient data to estimate calibration reliably.

**Exclusion of boundary artifacts.** At high abstention thresholds, samples near the threshold boundary (those that just barely abstain) have high value estimates and correspondingly high accuracy. Isotonic regression produces piecewise-constant fits; with finite data, if all samples in the highest fitted bin happen to be correct, the function plateaus at exactly 1. This is an artifact of fitting a piecewise-constant function to finite data at the boundary of the observed range, not a failure of the calibration methodology. We exclude points where $\hat{r}_\perp = 1$ because we cannot reliably distinguish $\hat{r}_\perp = 0.95$ from $\hat{r}_\perp = 1$ in this regime.

The x-axis range in Figure 3 thus reflects the achievable accuracy-abstention tradeoff: it displays precisely the region where we have valid empirical support for comparing methods.

## E. Threshold Calibration Across Data Splits

A practical concern is whether the threshold $T_\alpha$ — calibrated on a small held-out set to achieve a target abstention rate $\alpha$ — transfers reliably to new data. We verify this by randomly splitting the test set into two halves 20 times per seed: calibrating $T_\alpha$ on one half and measuring the achieved abstention rate on the other. Figure 7 plots achieved versus target abstention rate across all model–dataset pairs. The curves lie almost exactly on the diagonal, with mean absolute error below 1.2 percentage points in all settings, confirming that the threshold is straightforward to set in practice.

## F. Abstention Timing

Figure 8 shows the mean and median number of tokens generated before abstention as a function of abstention rate $\alpha$. At moderate abstention rates ($\alpha = 0.6$–$0.8$), abstention occurs at a mean of 16–34 tokens across settings, justifying $k = 20$ as a representative fixed-position baseline while illustrating that no single $k$ can match the dynamic method across all operating points.

Figure 9 shows the same quantity expressed as a fraction of the full trace length $c$. Abstention consistently occurs in the first half of generation: even at low abstention rates ($\alpha = 0.1$), the mean $\tau/c$ is below 0.5 across all settings. As $\alpha$ increases, abstention occurs progressively earlier, with mean $\tau/c$ below 0.15 at $\alpha = 0.9$. This confirms that the method identifies unpromising traces early and terminates them well before the full chain-of-thought is completed.

## G. Robustness of the Value Estimator

When the abstention threshold is set by targeting a desired abstention rate $\alpha$, the threshold is the $\alpha$-quantile of per-sample minimum trajectory values. Under this protocol, selective accuracy depends only on the *ranking* induced by $\hat{V}$, not on whether its values are calibrated probabilities. We verify two consequences of this.

**Invariance to monotone reparametrizations.** Applying a strictly monotone transformation $g$ to $\hat{V}$ leaves all rankings unchanged and therefore leaves selective accuracy exactly unchanged. Figure 10 confirms this: applying $g(v) = v^2$, $g(v) = \sqrt{v}$, and $g(v) = \sigma(5(v - 0.5))$ produces curves that overlap exactly across all settings.

**Degradation under additive noise.** We add Gaussian noise $\mathcal{N}(0, \sigma^2)$ to $\hat{V}$, expressing $\sigma$ as a multiple of the standard deviation of per-sample minimum trajectory values to make the scale comparable across datasets. Figure 11 shows selective accuracy as noise increases. GSM8K is highly robust ($\approx$1–2% drop at $\sigma = 1\times$std; $\approx$7–9% at $\sigma = 2\times$std). OlympiadBench shows higher sensitivity, reflecting that the harder task leaves less margin in the value function before perturbation disrupts rankings. In all settings the method retains meaningful gains over the no-abstention baseline at all noise levels tested.

## H. Further Implementation and Experimental Setup Details

**Experimental Setup:** We used an 80/20 train-test split with a fixed random seed of 42 for reproducibility across all experiments. The tokenwise value head method was trained for 3 epochs with learning rate $1 \times 10^{-4}$, batch size 8 (reduced to 2 for larger models), and AdamW optimizer with 0.1 dropout. The first-token baseline used identical hyperparameters and computed loss exclusively on the first output token. The LoRA abstention model employed 3 epochs, batch size 2, gradient accumulation over 10 steps, LoRA rank 16 with $\alpha = 32$, and targeted all projection layers (q_proj, v_proj, k_proj, o_proj, gate_proj, up_proj, down_proj). All methods used abstention thresholds of [0.3, 0.5, 0.7]. The datasets used RTP (Reason-Then-Predict) format where models generate reasoning followed by predictions, with correctness labels derived

from multiple model outputs rather than explicit self-verification queries.

**Hardware and Training Times:** All experiments were conducted on CUDA-enabled GPUs using mixed precision training (fp16 for most models, bf16 for Phi-3). Models were trained with frozen base parameters except for LoRA experiments. Training times varied significantly by method: the tokenwise value head required approximately 10-40 hours total (2-8 hours per epoch depending on batch size and hardware), the first-token baseline completed in 2-4 hours, and LoRA abstention training took 9-18 hours. We evaluated on multiple model families including Phi-3-small-8k-instruct, and Mistral-7B-Instruct-v0.3 across mathematical reasoning datasets (OlympiadMath, OlympiadPhysics, GSM8K).

## I. Improvement Over No Abstention as a Function of Abstention Rate

Figure 12 presents the magnitude of improvement in reward over no abstention, as a function of abstention rate $\alpha \in \{0.02, 0.04, ..., 0.98\}$. We see that estimated reward rises with the abstention rate, supporting the intuition behind Proposition 4.9.

## J. Abstention Precision

A complementary view of selective accuracy is the *precision* of the abstention decision: P(incorrect | abstained), the fraction of abstained samples that would have been answered incorrectly. A method with high precision selectively targets incorrect traces; a method abstaining uniformly at random achieves only the base error rate.

Figure 13 reports this metric across abstention rates for all methods, with the base error rate shown as a dashed reference. Dynamic abstention consistently exceeds both the base error rate and all baselines across all settings and abstention rates, for both models and both datasets. This confirms that abstentions are not chosen indiscriminately but are concentrated on traces the probe identifies as likely to fail.

## K. Fixed-Position Baseline Comparison

A natural baseline for dynamic abstention is a method that makes the abstention decision at a single fixed token position $k$, rather than dynamically. This corresponds to $f(\pi; k)$ from Definition 4.3 evaluated mid-generation. We compare against fixed-position versions of the constant step probe, LoRA abstention, and self-assessment baselines, all evaluated at the same position $k$ so that every fixed-position method benefits equally from observing $k$ tokens of partial generation. The MLP probe and the LoRA abstention head are retrained *specifically* at position $k$, rather than reusing models trained across all positions, ensuring a fair comparison.

**Arbitrariness of $k$.** A fundamental difficulty with fixed-position baselines is that there is no principled way to choose $k$: the optimal position varies across datasets, models, and desired abstention rates, and any single choice is inherently arbitrary. For mathematical reasoning, abstention rates of $60$–$80\%$ represent a natural operating regime, where the model answers only the questions it is most confident about and routes the remainder to a stronger solver or a human reviewer. At these abstention rates on GSM8K, the dynamic method abstains at a mean of $16$–$34$ tokens (median $9$–$14$ tokens) across Phi-3 and Qwen; this range shifts substantially with $\alpha$, so no fixed $k$ can match the dynamic method's operating point across all settings simultaneously. We choose $k = 20$ for GSM8K as a round number within this range, and $k = 100$ for OlympiadBench to reflect the substantially longer reasoning traces on harder problems. Both choices are arbitrary: a practitioner deploying a fixed-position baseline would face exactly this problem, with no data-independent criterion for selecting $k$. This arbitrariness is itself an argument for the dynamic approach.

**GSM8K results ($k = 20$).** All experiments in this section use $5$ seeds and report $\pm 1$ standard deviation. Even on GSM8K, the dataset for which our method shows the smallest gains over baselines in the main experiments, the dynamic method outperforms all fixed-position baselines at every abstention rate. The fixed-position methods plateau near the no-abstention accuracy regardless of $\alpha$. Specifically, at $\alpha = 0.7$, the best fixed-position baseline achieves $0.871$ on Phi-3 and $0.885$ on Qwen, while the dynamic method reaches $0.975$ and $0.990$, respectively. This gap widens as $\alpha$ increases, precisely because the dynamic method can adapt its stopping point to each trace while fixed-position methods cannot.

**OlympiadBench results ($k = 100$).** The dynamic method outperforms the fixed-position baseline across all abstention rates on both Phi-3 and Qwen. The one setting in which the fixed-position baseline shows meaningful improvement over

input-only methods is OlympiadBench with Qwen, where the longer traces give a fixed cutoff more opportunity to coincide with informative moments in the trajectory; even there, it remains substantially below the dynamic method. This is consistent with our theoretical analysis: fixed-position abstention can help when a single well-chosen $k$ aligns with the trajectory's informative moment, but cannot match a rule that adapts per-trace.

Full per-rate accuracy curves and tabulated results for both datasets are included in the supplementary material.

## L. Toxicity Avoidance Experiment Details

**Data.** We sample $500$ sentence beginnings uniformly at random from RealToxicityPrompts (Gehman et al., 2020) for training and an independent $500$ for test.

**Labeling.** For each sentence (beginning concatenated with model continuation), we query Claude with the template "`Sentence:  <full sentence>.  Is this sentence toxic?  Answer Yes/No.`" The binary non-toxicity label is $1$ if the answer is `No` and $0$ if `Yes`.

**Model and generation.** We use Qwen2.5-7B-Instruct with generation hyperparameters matching Section 6.1. Abstention triggers a fixed safe refusal string in place of continued generation.

**Baselines.** We compare against all three baselines from Section 6.1: Constant Step Probe, self-assessment, and LoRA abstention, all trained on the same $500$-prompt train set. Because RTP responses are sentence-length, fixed-position mid-generation variants are not meaningful; we report only input-processing ($t = 0$) variants.

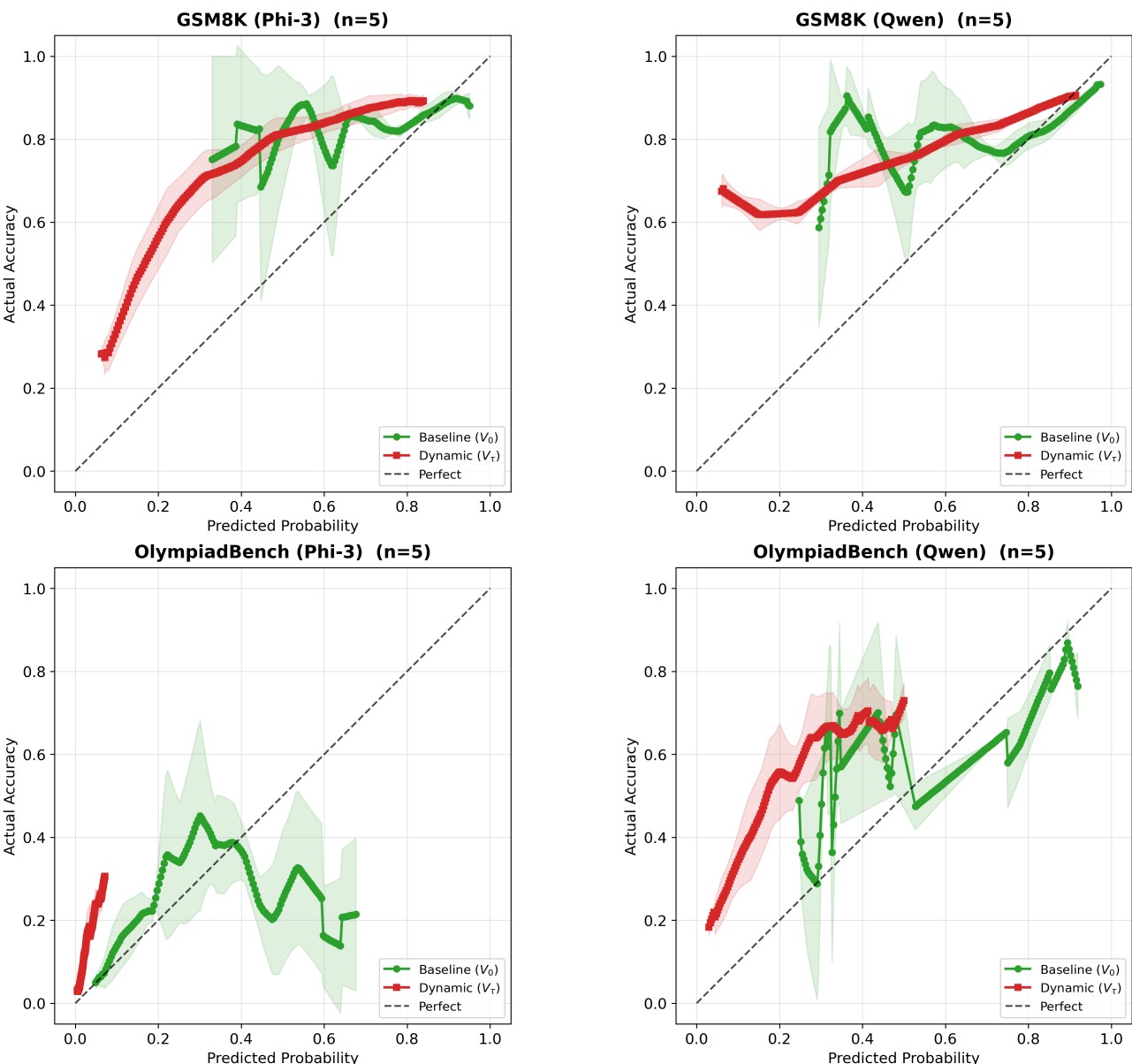

*Figure 6.* Calibration comparison between baseline (value at $t = 0$) and dynamic abstention (value at abstention time $\hat{V}_\tau$).

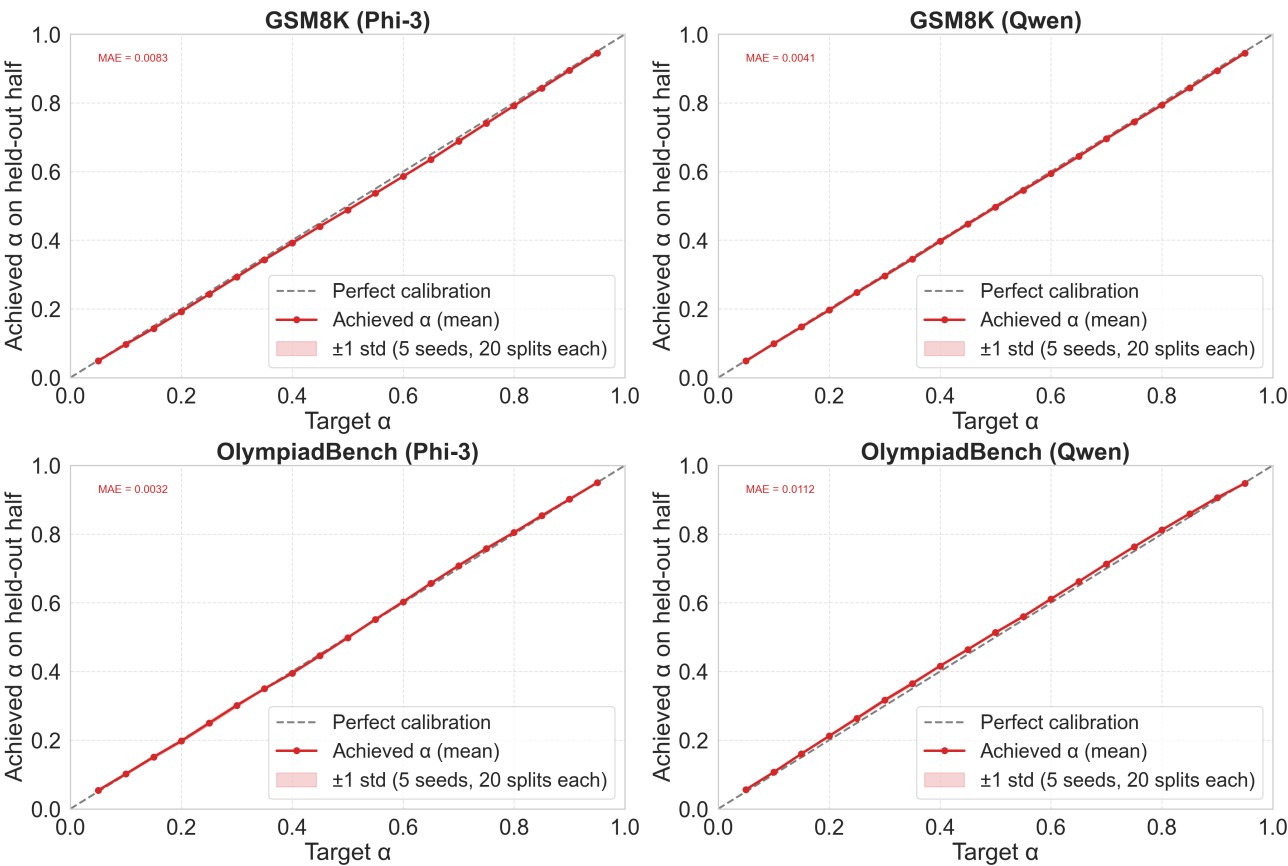

*Figure 7.* Achieved abstention rate on held-out split versus target abstention rate. Each curve is averaged over 5 seeds $\times$ 20 random splits; shaded regions show $\pm 1$ standard deviation. Mean absolute error (MAE) is annotated per panel.

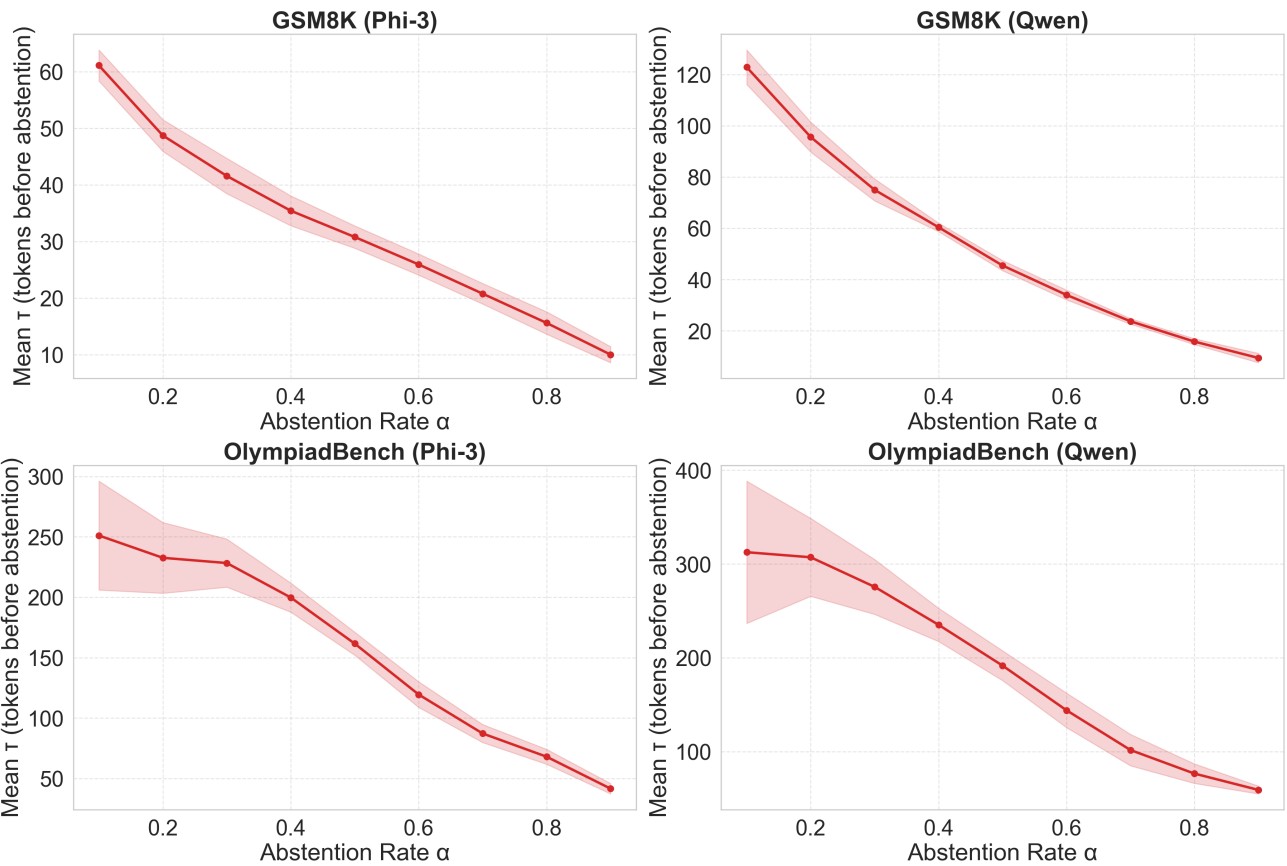

*Figure 8.* Mean and median tokens before abstention versus abstention rate. The range shifts substantially with $\alpha$, illustrating why no single fixed position $k$ can match the dynamic method across operating points.

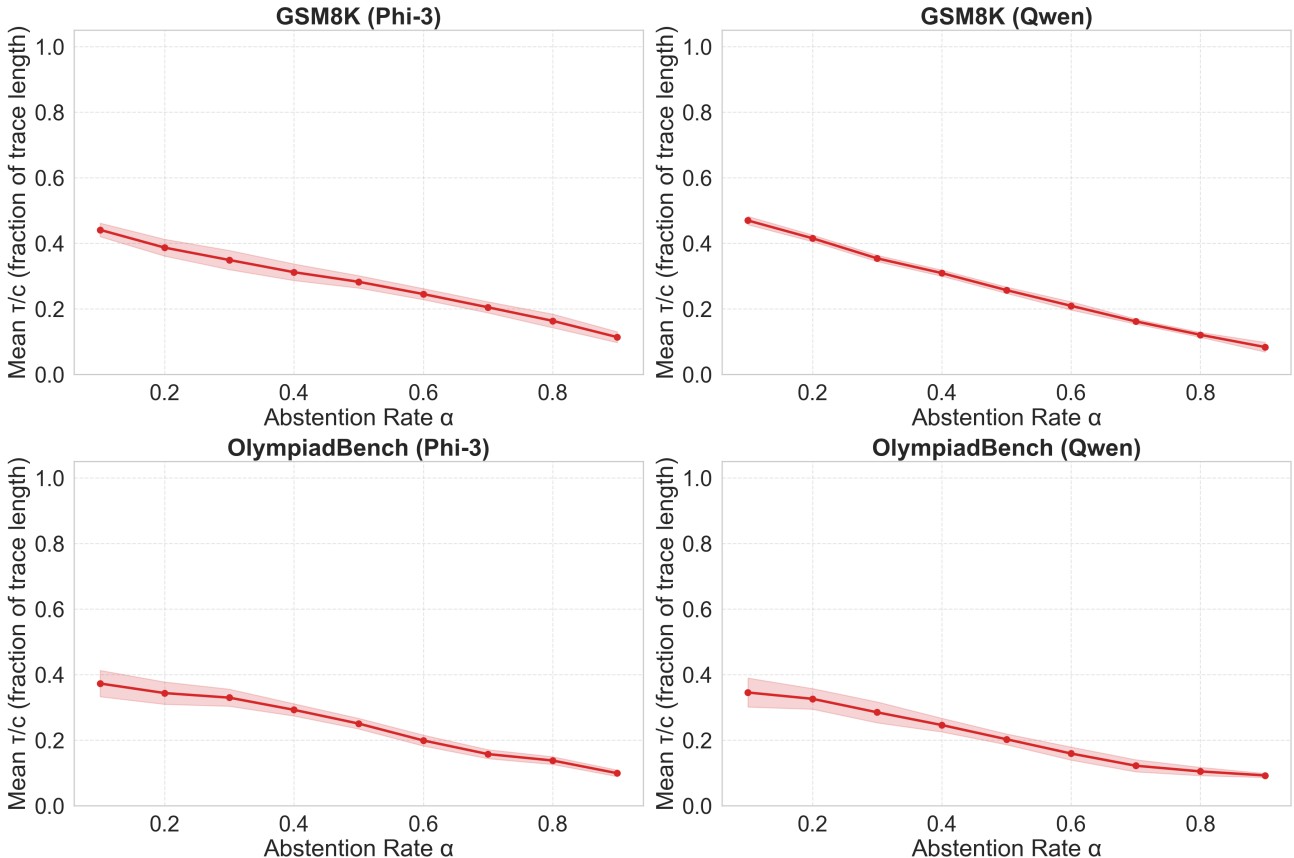

*Figure 9.* Mean abstention time $\tau$ as a fraction of full trace length $c$, versus abstention rate. Abstention consistently occurs in the first half of generation across all settings.

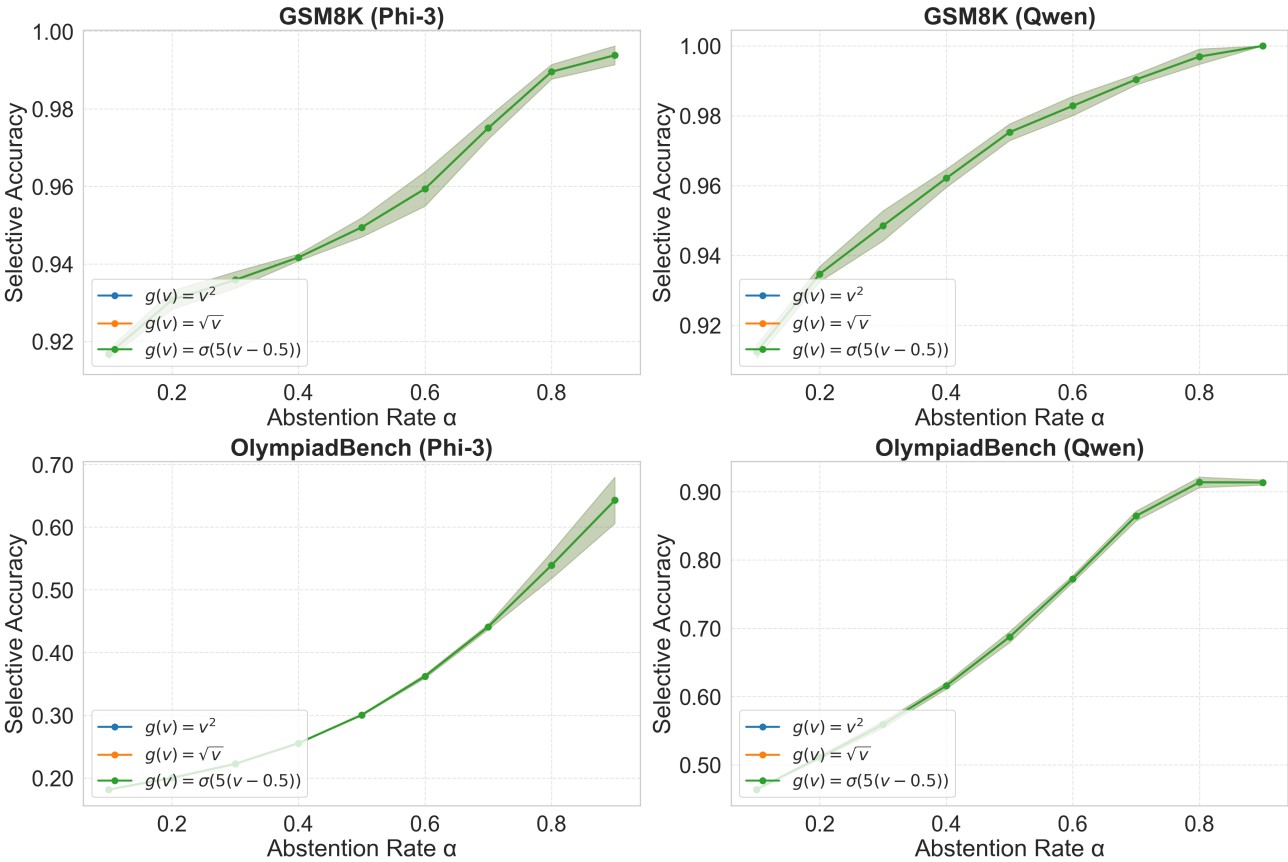

*Figure 10.* Selective accuracy under monotone reparametrizations of $\hat{V}$. All three transforms produce identical curves, confirming exact invariance.

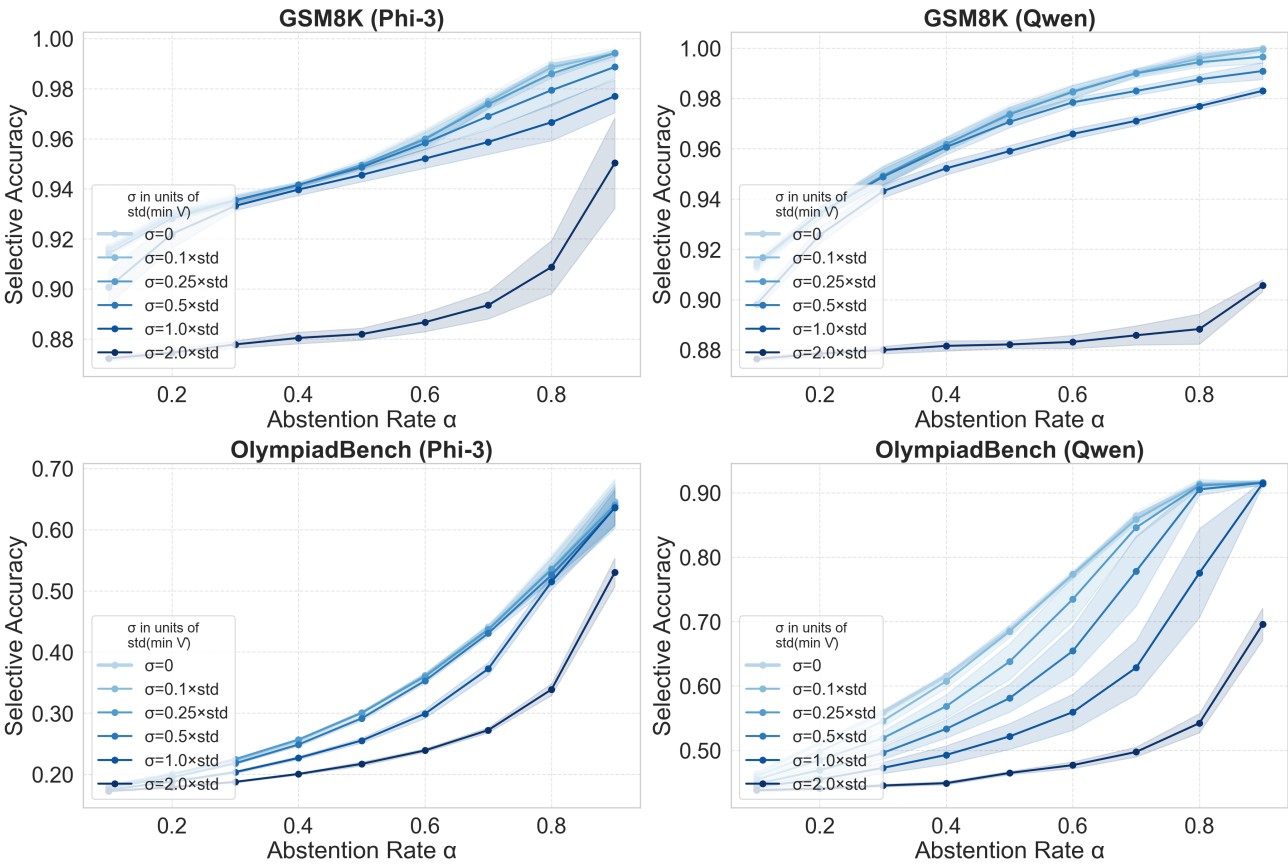

*Figure 11.* Selective accuracy under additive Gaussian noise to $\hat{V}$. Noise magnitude $\sigma$ is expressed in units of the standard deviation of per-sample minimum trajectory values. Performance degrades gracefully; gains over no-abstention are retained at all noise levels.

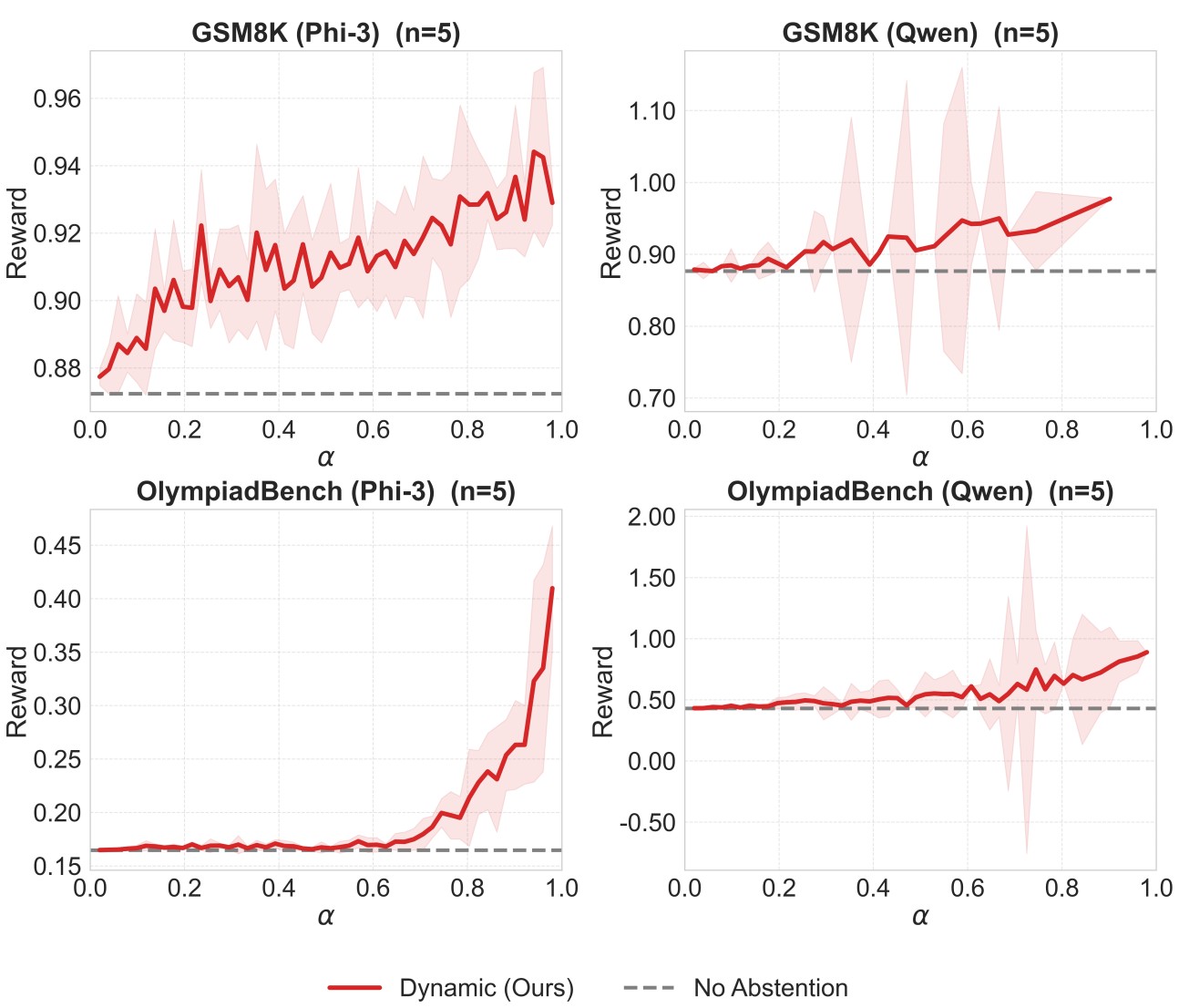

*Figure 12.* Estimated reward versus abstention rate.

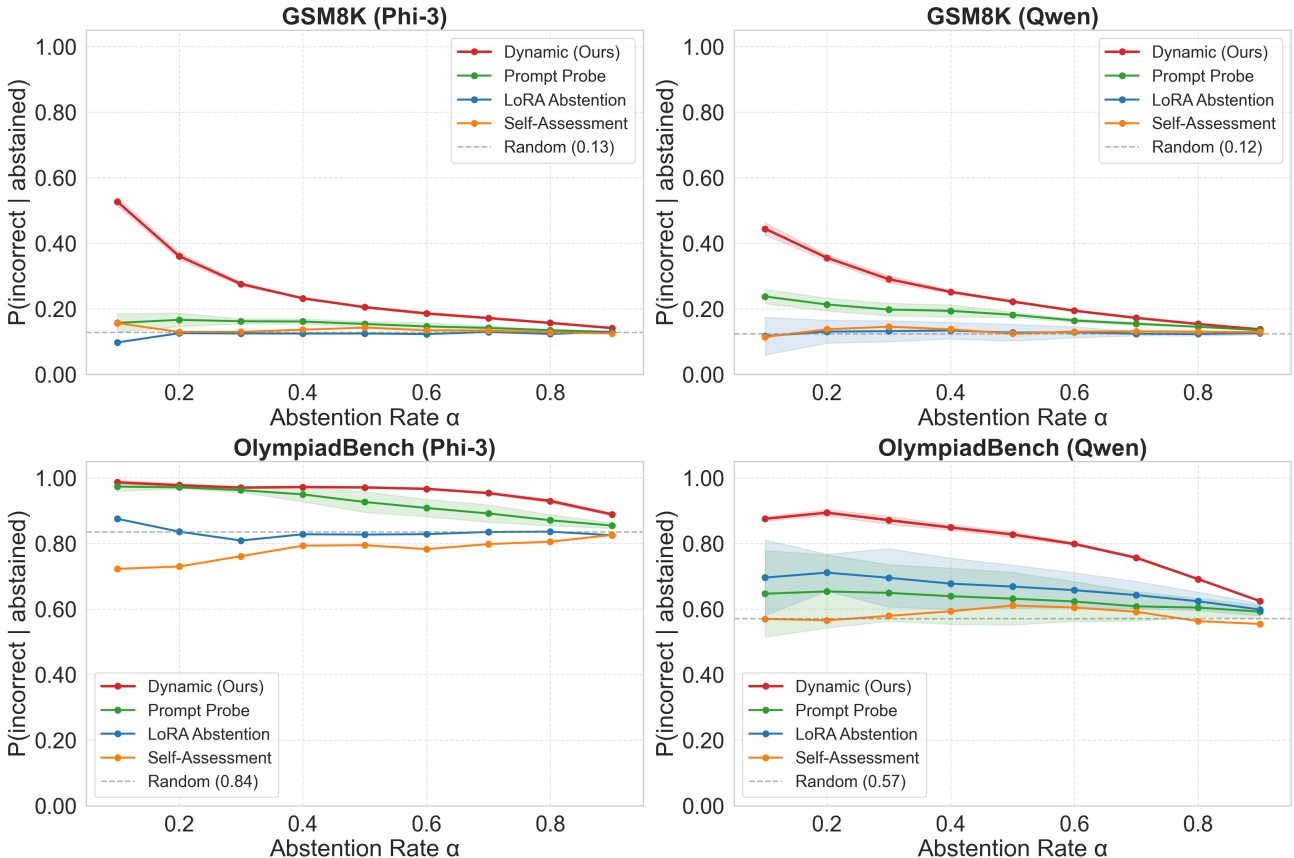

*Figure 13.* Precision of abstention: P(incorrect | abstained) versus abstention rate. The dashed line shows the base error rate (random abstention baseline). Dynamic abstention targets incorrect traces more precisely than all baselines across all settings.

