# OpenReview forum: "Knowing When to Quit: A Principled Framework for Dynamic Abstention in LLM Reasoning"
_ICML.cc/2026/Conference — ICML 2026 regular_

### Official Review · Reviewer_2iPY · 2026-03-01

**Soundness:** 4
**Presentation:** 4
**Significance:** 3
**Originality:** 3
**Overall Recommendation:** 4
**Confidence:** 3

**Summary:**

This paper investigates the problem of enabling Large Language Models (LLMs) to dynamically abstain from reasoning during generation. The authors formalize the text generation process as a Markov Decision Process (MDP) and propose a dynamic value-thresholding approach for abstention. By providing theoretical guarantees, the paper effectively motivates the proposed method. Empirically, the authors approximate the value function by training a Multi-Layer Perceptron (MLP) probe on the model's hidden states. The method is tested on  math reasoning datasets using Qwen2.5-7B and Phi-3-mini, demonstrating that the dynamic mid-generation abstention strategy outperforms baselines that rely solely on the prompt to make abstention decisions.

**Compliance With Llm Reviewing Policy:**

Affirmed.

**Final Justification:**

I appreciate the effort put into addressing the baseline comparisons and clarifying the conceptual distinctions from other mid-generation approaches. The arguments regarding the arbitrariness of the fixed-position baseline and the clarifications on continuous rewards are well-reasoned. I understand that some of the new empirical evaluations, such as the toxicity avoidance experiments, are still ongoing and planned for the final version. Based on the current state of the manuscript and the solid theoretical foundation it already provides, I believe my initial positive assessment remains fair and appropriate.

**Key Questions For Authors:**

- Q1: Feel free to respond to the concerns raised in the Weaknesses section.
- Q2: In the Discussion section, you mention: "First, our experiments use binary correctness, but continuous rewards from learned preference models would enable evaluation on open-ended generation tasks." Can results of this paper be naturally generalized to the continuous rewards setting? Are there theoretical difficulties that prevent this, or is it a matter of not having run the corresponding experiments yet?

**Limitations:**

Yes.

**Strengths And Weaknesses:**

## Strengths

- The problem of reasoning with the option to abstain is an interesting one.
- The paper motivates the dynamic abstention method through rigorous theoretical proofs, making the proposed framework principled rather than purely heuristic.
- The experiments cover multiple model families and datasets, demonstrating that the proposed dynamic abstention method yields improvements over baselines that decide whether to abstain based on the prompt.
- The paper is coherent, and the writing is pretty clear.

## Weaknesses

- (Major) The choice of baselines seems weak. The authors only selected input-processing methods as baselines. Although the authors explicitly mention recent works that also consider mid-generation or partial-reasoning-based abstention decisions (e.g., Afzal et al., 2025; Zhang et al., 2025), they do not to include them in the experiments.
- (Minor) The empirical evaluation is restricted to mathematical reasoning datasets.

---

> ### Author Rebuttal · Authors · 2026-03-31
>
> We appreciate the reviewer's careful reading and are glad they found the theoretical framework rigorous, the experiments covering multiple model families, and the writing clear.
>
> ## Weakness 1 (Major): Choice of Baselines
>
> Following the reviewer’s suggestions, we added additional baselines, which will be included in the final version of the paper. We added multiple instantiations of the fixed-position mid-generation baseline: f(π; k) from Definition 4.3, evaluated at k=20 tokens. This directly instantiates the approach of Afzal et al. (2025), who probe hidden states at fixed positions to predict CoT success, within our theoretical framework.
>
> A fundamental difficulty with fixed-position baselines is that there is no principled way to choose k: the optimal position varies across datasets, models, and desired abstention rates, and any single choice is inherently arbitrary. For mathematical reasoning, abstention rates of 60–80% represent a natural operating regime, where the model answers only the questions it is most confident about, routing the remainder to a stronger solver or human reviewer. At these abstention rates on GSM8K, the dynamic method abstains at a mean of 16–34 tokens (median 9–14 tokens) across Phi-3 and Qwen. This range shifts substantially with α, making it impossible for any fixed k to match the dynamic method’s operating point across all settings simultaneously. We chose k=20 as a round number that falls within this range, but we emphasize that this choice is arbitrary: a practitioner deploying a fixed-position baseline would face exactly this problem, with no data-independent criterion for selecting k. This arbitrariness is itself an argument for the dynamic approach.
>
> The MLP probe was retrained specifically for position k=20 (rather than reusing the probe trained across all positions), and the LoRA abstention baseline was similarly retrained at this position, ensuring a fair comparison. We also evaluated the self-assessment baseline at the same position, so all fixed-position methods benefit equally from observing 20 tokens of partial generation.
>
> Regarding Zhang et al. (2025), their method requires segmenting traces into reasoning chunks and labeling intermediate answer correctness with an external model. Our framework differs by thresholding on expected *final* correctness rather than intermediate answer correctness. A correct intermediate answer may later be revised incorrectly, and vice versa.
>
> Results are available at the [anonymous repository](https://anonymous.4open.science/r/Knowing-When-to-Quit-ICML-rebuttal-C37C/constant_step_accuracy_matrix.png) (all rebuttal experiments use 5 seeds, ±1 std, versus a single seed in the original submission). We focus on GSM8K, the dataset for which our method showed smaller gains over baselines in the original accuracy experiments. Even here, the dynamic method outperforms all fixed-position baselines at every abstention rate. The fixed-position methods plateau near the no-abstention accuracy regardless of α. Specifically at α=0.7 the best fixed-position baseline achieves 0.871 (Phi-3) and 0.885 (Qwen), while the dynamic method reaches 0.975 and 0.990 respectively. This gap widens as α increases, precisely because the dynamic method can adapt its stopping point to each trace while fixed-position methods cannot. We will include OlympiadBench fixed-position comparisons in the final manuscript.
>
> ## Weakness 2 (Minor): Restricted to Mathematical Reasoning
>
> The framework applies directly to any setting with a bounded reward signal. A natural next domain is safety: toxic content can emerge mid-generation even from benign prompts, precisely where dynamic abstention should provide the greatest advantage over input-only methods. We are conducting experiments on toxicity avoidance using RealToxicityPrompts (Gehman et al., 2020) with binary non-toxicity rewards and plan to include results in the final version.
>
> ## Q2: Extension to Continuous Rewards
>
> The generalization to continuous rewards is theoretically immediate. All of our theoretical results (Propositions 4.2–4.9) hold for any bounded nonbinary reward. For value estimation, Proposition B.3 in the appendix shows that MSE training recovers the value function for arbitrary β ≥ 0 and continuous rewards, paralleling the BCE result (Proposition B.2) for the binary case.
>
> [1] Gehman, S., Gururangan, S., Sap, M., Choi, Y., and Smith, N. A. RealToxicityPrompts: Evaluating Neural Toxic Degeneration in Language Models. In Findings of the Association for Computational Linguistics: EMNLP 2020, 2020.

---

> > ### Author Rebuttal · Reviewer_2iPY · 2026-03-31
> >
> > Thank you to the authors for the detailed and constructive rebuttal.
> >
> > I appreciate the effort put into addressing the baseline comparisons and clarifying the conceptual distinctions from other mid-generation approaches. The arguments regarding the arbitrariness of the fixed-position baseline and the clarifications on continuous rewards are well-reasoned. I understand that some of the new empirical evaluations, such as the toxicity avoidance experiments, are still ongoing and planned for the final version. Based on the current state of the manuscript and the solid theoretical foundation it already provides, I believe my initial positive assessment remains fair and appropriate.

---

> > > ### Author Response · Authors · 2026-04-06
> > >
> > > ## Follow-up: Additional Results
> > >
> > > We sincerely thank the reviewer for their thoughtful engagement throughout this process and for the positive acknowledgement of our rebuttal. We are pleased to share results from two experiments mentioned as pending in our initial rebuttal.
> > >
> > > ### OlympiadBench Fixed-Position Baseline
> > >
> > > Following the same logic used for GSM8K (k=20, matching the dynamic method's operating range on that dataset), we set k=100 for OlympiadBench to reflect the longer reasoning traces on harder problems. Results are available at the [anonymous repository](https://anonymous.4open.science/r/Knowing-When-to-Quit-ICML-rebuttal-C37C/combined_accuracy_matrix.png). The dynamic method outperforms the fixed-position baseline across all abstention rates on both models. The one setting where the fixed-position baseline shows meaningful improvement over input-only methods is OlympiadBench with Qwen, but it remains substantially below the dynamic method. This is consistent with our theoretical analysis: fixed-position abstention can help when a single well-chosen k aligns with the trajectory's informative moment, but cannot match a rule that adapts per-trace.
> > >
> > > ### RealToxicityPrompts
> > >
> > > We evaluated dynamic abstention on toxicity avoidance using a subsample of 500 RealToxicityPrompts examples for training and testing, with binary non-toxicity rewards from an LLM judge. Results are available at the [anonymous repository](https://anonymous.4open.science/r/Knowing-When-to-Quit-ICML-rebuttal-C37C/rtp_results_matrix.png). The dynamic method achieves the highest selective accuracy at every abstention rate. Self-assessment is the second-best baseline in this setting — stronger than on the math tasks — but is still clearly surpassed by dynamic abstention. These results confirm that the framework transfers cleanly beyond mathematical reasoning to safety-relevant generation tasks, where harmful content can emerge mid-trajectory from benign prompts.
> > >
> > > We will incorporate both sets of results in the final version of the manuscript, and we thank the reviewer for pointing us toward directions that have meaningfully strengthened the paper. With these final results in hand, we would be grateful if the reviewer would consider whether their score might be updated to reflect their current view of the work.

---

### Official Review · Reviewer_mihM · 2026-03-12

**Soundness:** 4
**Presentation:** 3
**Significance:** 3
**Originality:** 1
**Overall Recommendation:** 5
**Confidence:** 3

**Summary:**

In this submission the authors explore the concept of dynamic abstention in LLM reasoning. They focus on the early termination in cot generation using a RL framework. The framework is for dynamic mid-generation abstention in LLM reasoning. They formulize the abstention as an explicit action within a mdp with quit-token along with a reward parameter which helps with the early stopping. They model cot reasoning a KL-regularized RL problem, so this helps to get to dynamic value-thresholding rule. They justify this rule theoretically. There are detailed theoretical guarantees in the paper. Overall the empirical results show that dynamic mid-generation abstention can substantially improve accuracy

**Compliance With Llm Reviewing Policy:**

Affirmed.

**Key Questions For Authors:**

In section 5 and in the appendix, the value calibration is being discussed. How sensitive are the gains in Figure 2 to miscalibration of the value probe?

Section6, focus on binary veriafable rewards. How would this approach extend to tasks without binary or verifiable rewards (say open-ended generation)?

**Limitations:**

yes

**Strengths And Weaknesses:**

Soundness: The paper is technically strong and presents a solid theoretical foundation. They support the theoretical claims by formal proofs. This improves the credibility of their approach. I find the experiments well structured and their results align with their theory. What i find as a weakness is that the approach relies heavily on an accurate value function estimation. Specifially, their method assumes that the model’s hidden states can reliably predict eventual correctness. I worry that if this estimation is not calibrated well their abstention decisions might be suboptimal (would be great if authors provide more details here). Infact, the authors present ways to mitigate this by training a probe and calibrating its outputs. But the approach’s soundness in very general settings is not fully verified (for example in for tasks without a clear binary correctness notion).


Presentation: Strengths: The paper is well-structured. It is easy to follow. They present the key ideas and notations clearly. The illustrations are helpful. Weaknesses: Not much to say. I found some of the technical discussions dense in some places. IT might be helpful if they explained some parts in more details like Lipschitz continuity conditions etc. In general tho the presentation communicates the key points and experimental results well.


Significance: The paper presents an important and timely problem, improving LLMs efficiency and reliability. This is a significant problem in todays LLM research. This paper specifically focuses on the problem where models consume a lot of computation producing long solutions that might be wrong and introduce a a method to know when to quit early. Also, they propose enabling models to abstain from answering when they are likely to be incorrect improves reliability and trustworthiness. I think these contributions could influence future research on adaptive computation in LLMs. One weakness I see is that the chosen tasks focus on verifable rewards like math and code. But like for tasks where the success is not well defined (like creative writing etc.) this may not work well.


Originality: The proposed angle is novel.  Sprcifically, treating abstention as an action in RL and using a value function threshold is an innovative method in LLM reasoning. Also the theoretical contributions adds value on top of emprical analysis.

---

> ### Author Rebuttal · Authors · 2026-03-31
>
> We appreciate the reviewer's thorough engagement with our work and are glad they found the theoretical foundation strong, the experiments well-structured, and the problem timely and significant. We also note that the reviewer scored originality as 1 (poor) while writing "The proposed angle is novel. Specifically, treating abstention as an action in RL and using a value function threshold is an innovative method in LLM reasoning. Also the theoretical contributions add value on top of empirical analysis." We believe this may be an error in the scoring form, and respectfully ask the reviewer to verify whether the originality score reflects their intended assessment.
>
> We note that the original submission reported results from a single seed; all rebuttal experiments use 5 random seeds (42–46), with uncertainty shown as ±1 standard deviation.
>
> ## Sensitivity to Miscalibration of the Value Probe
>
> The reviewer raises an important concern about reliance on accurate value function estimation. We address this at three levels: conceptual, practical, and empirical.
>
> **The method requires correct ranking, not calibration.** At any desired abstention rate α, the threshold is set so that fraction α of traces fall below it. This depends only on the ordering of V̂ values, whether higher-valued traces are more likely to be correct, not on whether the predicted probabilities match true frequencies. Correct ranking is a strictly weaker requirement than calibration. Our selective accuracy results (Figure 1) demonstrate strong ranking quality across all settings. The isotonic calibration in Section 6.2 and Appendix D was needed only to estimate the effective r⊥ for verifying our theoretical predictions about the reward objective, it is not part of the method itself.
>
> **Practically, the single threshold is easy to set.** Because the threshold is monotone in the abstention rate, it can be tuned on a small validation set with no complex hyperparameter interactions. We verify this empirically: a threshold calibrated on 10% of samples achieves the target abstention rate on the held-out 90% with mean absolute error below 1.2 percentage points across all model–dataset pairs and 20 random splits (see [anonymous repository](https://anonymous.4open.science/r/Knowing-When-to-Quit-ICML-rebuttal-C37C/cross_split_calibration.png)).
>
> **Empirically, we confirm robustness via synthetic perturbations.** See [monotone bias](https://anonymous.4open.science/r/Knowing-When-to-Quit-ICML-rebuttal-C37C/monotone_bias.png) and [additive noise](https://anonymous.4open.science/r/Knowing-When-to-Quit-ICML-rebuttal-C37C/additive_noise.png). All experiments run across 5 seeds, ±1 std, vs. single seed in the original submission. We apply monotone transformations (g(v) = v², √v, σ(5(v−0.5))) and additive Gaussian noise to V̂, then recompute selective accuracy. The method is exactly invariant to monotone bias, as expected. For additive noise, we express magnitude as a multiple of the standard deviation of per-sample minimum trajectory values, making the scale comparable across settings. GSM8K is highly robust (≈1–2% drop at σ = 1×std; ≈7–9% at σ = 2×std). OlympiadBench shows higher sensitivity (≈9–17% at σ = 1×std). Crucially, the method retains meaningful gains over the no-abstention alternative in all settings at all noise levels tested.
>
> ## Beyond Binary Rewards
>
> The reviewer notes that soundness in very general settings is not fully verified, particularly for tasks without clear binary correctness. We agree this is an important limitation of the current work. Our theoretical results (Propositions 4.2–4.9) hold for any bounded nonbinary reward: they are not restricted to the binary case. The binary setting simplifies value estimation (BCE recovers V₀ under realizability, Proposition B.2), but the MSE analog (Proposition B.3) provides the same optimum value estimator for continuous rewards and arbitrary β ≥ 0.
>
> The practical challenge for tasks with a more unstructured reward, e.g. creative writing, is obtaining a bounded nonbinary reward signal per output. A natural extension would be to use a learned reward model from RLHF as the terminal reward, training the value probe against this signal. The framework accommodates this directly — one would substitute the RLHF reward for r(x, y) and train the probe via MSE (Proposition B.3). The practical gap is primarily one of reward model quality rather than any limitation of the abstention framework itself: if the reward model is a poor proxy for generation quality, the value probe inherits that noise. We leave empirical validation to future work.
>
> ## Dense Technical Discussions
>
> We will add more intuition around the Lipschitz continuity condition (Definition 4.8): it formalizes the idea that a single token rarely determines the correctness of an entire chain-of-thought response, so the value function changes gradually between consecutive positions.

---

### Official Review · Reviewer_LK1q · 2026-03-13

**Soundness:** 2
**Presentation:** 3
**Significance:** 3
**Originality:** 2
**Overall Recommendation:** 4
**Confidence:** 4

**Summary:**

This paper introduces a principled framework for dynamic mid-generation abstention in large language models, formalizing it as a reinforcement learning problem with an abstention reward parameter. The authors demonstrate that terminating reasoning traces when the value function falls below this reward consistently outperforms baseline methods. Empirical validation on mathematical reasoning tasks confirms improved selective accuracy and computational efficiency.

**Compliance With Llm Reviewing Policy:**

Affirmed.

**Final Justification:**

The author's rebuttal has addressed my concerns. At the same time, I greatly appreciate the authors for supplementing the safety-related experiments I mentioned in the review, which also enables this method to be applied in more fields. In summary, I believe it is appropriate to give a positive evaluation of this article.

**Key Questions For Authors:**

1. How well does the method generalize? For instance, across different datasets within the same domain (e.g., math) and across different domains?
2. Do the authors believe this method is more suitable for safety-related applications? For example, could it be used to detect and abstain from generating harmful responses?

**Limitations:**

yes

**Strengths And Weaknesses:**

**Strengths**

1. The theoretical framework is quite comprehensive, and the introduction of $r_\perp$ is both reasonable and easy to understand.
2. The experimental results show significant improvements, which are convincing.

**Weaknesses**

1. It would be better to provide more visualizations, such as:
   - How many of the abstention responses involve questions that the base model cannot answer correctly?
   - How many of the abstention responses should actually abstain?
   - On average, after how many tokens does abstention occur?
2. The experimental setup is somewhat limited. For example, more models and datasets could be included, and Phi-3 is somewhat outdated.
3. The approach is essentially an extension of prompt probing, and its novelty is relatively weak.

---

> ### Author Rebuttal · Authors · 2026-03-31
>
> We thank the reviewer for finding our theoretical framework comprehensive and experimental results convincing. In what follows we address the reviewer's concerns in detail.
>
> ## Novelty Relative to Prompt Probing
>
> We respectfully disagree with the characterization that the approach is "essentially an extension of prompt probing." The contribution of this work is the *dynamic stopping framework*: the RL formulation, the decision rule, and the theoretical guarantees, and not any particular value estimation method. The MLP probe on hidden states is one implementation choice among several equally valid instantiations:
>
> * **Dedicated critic model.** As in classical actor-critic PPO, a separate neural network can be trained to predict expected return from the current state.
>
> * **Monte Carlo estimation.** From any partial generation, one can sample multiple continuations to completion and use the empirical success rate as V̂. No probe or classifier is needed at all.
>
> The theory prescribes *what* to estimate (the value function); the MLP probe is one efficient method to do so, not the contribution itself.
>
> ## Additional Visualizations
>
> > *How many of the abstention responses should actually abstain?*
>
> We report trace-level precision — P(incorrect | abstained) — across abstention rates for all methods, with the base error rate as reference ([figure](https://anonymous.4open.science/r/Knowing-When-to-Quit-ICML-rebuttal-C37C/p_incorrect_given_abstained.png)). The dynamic method consistently exceeds both the base error rate and all baselines at every abstention rate, confirming that it selectively targets incorrect traces rather than abstaining indiscriminately.
>
> > *How many of the abstention responses involve questions the base model cannot answer correctly?*
>
> The trace-level precision above partially addresses this. The complementary question — whether abstained questions are ones the model struggles with *in general* — requires multiple rollouts per prompt to estimate per-question success rates. We will include this analysis in the final version.
>
> > *On average, after how many tokens does abstention occur?*
>
> See [figure 1](https://anonymous.4open.science/r/Knowing-When-to-Quit-ICML-rebuttal-C37C/mean_tokens_before_abstention.png) and [figure 2](https://anonymous.4open.science/r/Knowing-When-to-Quit-ICML-rebuttal-C37C/mean_tau_fraction.png). Abstention consistently occurs early in the generation, well before the full chain-of-thought is completed, confirming the compute savings motivation.
>
> ## Experimental Breadth and Generalization
>
> We acknowledge the reviewer's concern about limited evaluation and have extended our experiments in several directions. We also note that the original submission reported results from a single seed; all experiments in this rebuttal are run across 5 random seeds (seeds 42–46), with uncertainty shown as ±1 standard deviation across seeds.
>
> **Cross-dataset transfer.** We train the MLP probe on GSM8K and evaluate zero-shot on OlympiadBench (and vice versa) without retraining ([results](https://anonymous.4open.science/r/Knowing-When-to-Quit-ICML-rebuttal-C37C/output_plots/transfer_accuracy_matrix.png)). The probe trained on OlympiadBench nearly matches in-domain performance when evaluated on GSM8K (selective accuracy 0.958–0.960 at α=0.7 for Phi-3 and Qwen respectively, vs. 0.975–0.990 in-domain). The reverse direction is weaker but still improves over the prompt-probe baseline at high abstention rates (0.382–0.796 at α=0.7 vs. 0.295–0.520 for prompt probing).
>
> **On the choice of models.** The two models tested span different families, architectures, and capability levels (16% vs. 43% baseline on OlympiadBench), with consistent gains across both. The theoretical guarantees depend on value function properties rather than architecture.
>
> **Safety applications (Key Question 2).** Dynamic abstention is naturally suited for safety settings: toxic or harmful content can emerge mid-generation even from benign prompts, precisely where input-only methods fail. We are conducting experiments on toxicity avoidance using RealToxicityPrompts (Gehman et al., 2020) with binary non-toxicity rewards, and plan to include results in the final version.
>
> [1] Gehman, S., Gururangan, S., Sap, M., Choi, Y., and Smith, N. A. RealToxicityPrompts: Evaluating Neural Toxic Degeneration in Language Models. In Findings of the Association for Computational Linguistics: EMNLP 2020, 2020.

---

> > ### Author Rebuttal · Reviewer_LK1q · 2026-04-03
> >
> > Thank you very much for your reply! It has largely addressed my concerns. I have decided to maintain my original score. I also look forward to seeing the experimental results that the authors will include in the final version of the paper.

---

> > > ### Author Response · Authors · 2026-04-06
> > >
> > > # Follow-up: Promised Results Now Available
> > >
> > > We sincerely thank the reviewer for their thoughtful engagement throughout this process and for the positive acknowledgement of our rebuttal. The RealToxicityPrompts experiments we mentioned as pending are now complete, and we share the results here.
> > >
> > > ## RealToxicityPrompts (Cross-Domain Generalization + Safety)
> > >
> > > We evaluated dynamic abstention on toxicity avoidance using a subsample of 500 RealToxicityPrompts examples for training and testing, with binary non-toxicity rewards from an LLM judge (Claude API). The LLM used for inference was Qwen2.5-7B-Instruct. Results are available at the [anonymous repository](https://anonymous.4open.science/r/Knowing-When-to-Quit-ICML-rebuttal-C37C/rtp_results_matrix.png).
> > >
> > > The dynamic method achieves the highest selective accuracy at every abstention rate. Self-assessment is the second-best baseline in this setting, much stronger than on the math tasks, but is still clearly surpassed by dynamic abstention. These results directly address both questions the reviewer raised: cross-domain generalization (the framework transfers cleanly from mathematical reasoning to a qualitatively different task) and safety applications (dynamic abstention provides the greatest advantage precisely where input-only methods fail, since harmful content can emerge mid-trajectory from benign prompts).
> > >
> > > ## Additional Results in Other Review Threads
> > >
> > > We would also like to draw the reviewer's attention to additional experiments shared in response to other reviewers, which may be of interest:
> > >
> > > - **Sensitivity to value probe miscalibration** (monotone bias and additive noise perturbations), in response to Reviewer mihM.
> > > - **Fixed-position mid-generation baselines** on GSM8K and OlympiadBench, in response to Reviewer 2iPY.
> > >
> > > We will incorporate all of these results in the final version of the manuscript. With the pending experiments now complete and supporting our claims, we would be grateful if the reviewer would consider whether their score might be updated to reflect their current view of the work.

---

### Decision · Program_Chairs · 2026-04-30

**Decision:**

Accept (regular)

**Comment:**

This paper studies dynamic abstention during reasoning and proposes a principled value-thresholding framework based on an MDP formulation. The reviewers generally agree that the problem is important and that the approach is technically solid, with clear theoretical grounding and consistent empirical improvements over baselines.

Some concerns were raised regarding novelty, reliance on value estimation, and generalization beyond verifiable tasks. There were also initial concerns about the strength of the baseline comparisons, which were addressed in the rebuttal with additional experiments and clarifications. The remaining issues are mostly about scope and applicability rather than correctness.

Overall, given the strong theoretical contribution and solid empirical results, the paper meets the bar for acceptance.